# Universal Black-Box Targeted Reward Poisoning Attack Against Online Deep Reinforcement Learning

**Yinglun Xu**                                                         *yinglun6@illinois.edu*
*Department of Computer Science*
*University of Illinois at Urbana-Champaign*

**Gagandeep Singh**                                                    *ggnds@illinois.edu*
*Department of Computer Science*
*University of Illinois at Urbana-Champaign*

**Reviewed on OpenReview:** *https://openreview.net/forum?id=MXOaDKu8lY*

## Abstract

This work proposes the first universal black-box targeted attack against online reinforcement learning through reward poisoning during training time. The attack is universal in that it can effectively mislead any efficient learning algorithms to follow targeted behaviors specified by the attacker under general assumptions. More specifically, we generalize a common feature of the efficient learning algorithms and assume that such algorithms would mostly take the optimal actions or actions close to them during training. We quantify the efficiency of an attack and propose an attack framework where it is feasible to evaluate the efficiency of any attack instance in the framework based on the assumption. Finally, we find an instance in the framework that requires a minimal per-step perturbation budget, which we call 'adaptive target attack.' We theoretically analyze and prove a lower bound for the attack efficiency of our attack in the general RL setting. Empirically, on a diverse set of popular DRL environments learned by state-of-the-art DRL algorithms, we verify that our attack efficiently leads the learning agent to various target policies with limited budgets.

## 1 Introduction

Online deep reinforcement learning (DRL) algorithms have great potential to be applied in industrial applications such as robotics (Christiano et al., 2017), recommendation systems (Afsar et al., 2021), and natural language processing Ouyang et al. (2022). In such applications, the reward signals during training are usually feedback from human users, which raises the threat of training-time reward poisoning attacks. There may exist adversarial human users who deliberately provide malicious rewards. A malicious attack goal can be manipulating the DRL agent to follow certain behaviors during training with malicious properties, such as unsafe or harmful behavior, to benefit the attacker. Such behavior can still lead to high performance according to the agent's metric, making the attack harder to detect. For example, consider the scenario of training a restaurant's recommendation system. A restaurant wants to mislead the recommendation system to recommend itself more likely. For this purpose, the restaurant can deliberately provide false negative feedback for its competitors and positive feedback for itself. Then, during training, a vulnerable recommendation system might mostly recommend the malicious restaurant, which is unfair and harms the benefits of the restaurants, users, and the system. We show the framework of targeted poisoning attacks against online RL in Figure 1. Prior studies on targeted attacks mainly focus on simpler tabular settings (Rakhsha et al., 2020; Xu et al., 2021; Zhang et al., 2020; Banihashem et al., 2022). The results only show the vulnerability of current tabular RL algorithms, and the attacks usually only work in the white-box setting against specific learning algorithms, which may not be practical. In this work, we aim to expose the practical vulnerabilities of state-of-the-art DRL algorithms by finding a targeted attack that works efficiently in a realistic setting.

**Universal black-box targeted attack in DRL and the challenges:** Here, we discuss the practical constraints on an attack.

1. **Universal black box:** The learning agent can directly defend against white-box attacks by hiding its learning algorithm and training parameters from the public. Given that many different efficient RL algorithms have been developed (van Hasselt et al., 2016; Wang et al., 2016; Haarnoja et al., 2018; Dankwa & Zheng, 2019), the agent can arbitrarily choose an efficient algorithm. Therefore, a realistic attacker should be oblivious to the learning agent (black-box attack), and its attack strategy should work against any efficient learning algorithms an agent might use (universal attack).

2. **Real-time with limited computational power:** In online RL, the data collection during the training process happens in real-time. Therefore, a realistic attack should also be able to inject perturbations during training in real-time. In addition, it is more practical to run the attack with limited computational resources, as rich computational resources may not always be available to a malicious attacker. We do not consider the computational cost of constructing the target policy, as it is an input to the targeted attack problem.

We aim at developing an attack that can mislead the agent to take target actions with strict constraints and budgets. We formally define the attack constraints and budgets in Section 2. Particularly, we consider optimizing the attack efficiency that is measured by how much attack budgets are required to successfully mislead the learning agent. We formally define the attack efficiency in Section 3. Developing an efficient attack is challenging. First, a universal attack should work for different algorithms an agent might use, and it is hard to determine exactly what these algorithms are for an agent. Second, in the black-box setting, the attacker cannot exactly predict an agent's behavior without detailed knowledge of the training setup. To achieve high efficiency, the attack should optimize the budget it needs during the training process while working with limited computational resources. This becomes especially difficult without predicting the agent's behavior under attack.

**Our contributions:** To the best of our knowledge, our attack is the first universal black-box targeted attack applicable to the DRL setting. The key insight for making the attack universal is a general assumption based on a feature of efficient learning algorithms: an efficient DRL algorithm can learn the optimal solution or a similar one through strategic exploration in the environment. Under the assumption, we characterize the set of efficient algorithms a rational learning agent might consider. In the universal black box setting, it is hard to approximate the behavior of all efficient learning algorithms under an arbitrary attack. We develop an attack framework called 'efficient adversarial reward engineering' such that the behavior of an arbitrary efficient learning algorithm becomes predictable under any attack under the framework. In this framework, the attacker perturbs the feedback signals during training as if they are generated by an adversarial RL environment desired by the attacker. Finally, we find the instance in the framework that requires a minimal attack budget on per-step perturbation and call it the 'adaptive target attack.' We provide a detailed theoretical analysis of the attack efficiency in the general RL setting and conduct complementary experiments in the DRL setting. The empirical results verify that our attack can successfully mislead the agent to a target policy with limited budgets and computational resources. Our attack works for general RL problems with both discrete and continuous action spaces and applies to diverse learning algorithms. We experimentally verify the efficiency of our attack on various popular DRL environments for continuous and classical control problems, including HalfCheetah, Hopper, Walker, and Acrobot, and various state-of-the-art DRL algorithms, including SAC (Haarnoja et al., 2018), TD3 (Dankwa & Zheng, 2019), DDPG (Lillicrap et al., 2015), and DQN (van Hasselt et al., 2016). We consider various target policies to show that our attack works in a wide range of practical scenarios. Among all the attack settings we consider, our results show that our black-box attack can always successfully make an agent take actions close to target actions during training with very limited budgets. As the target actions are also the optimal actions in the adversarial RL environment created by the attacker, the results also indirectly verify the assumption we made on the efficient learning algorithms.

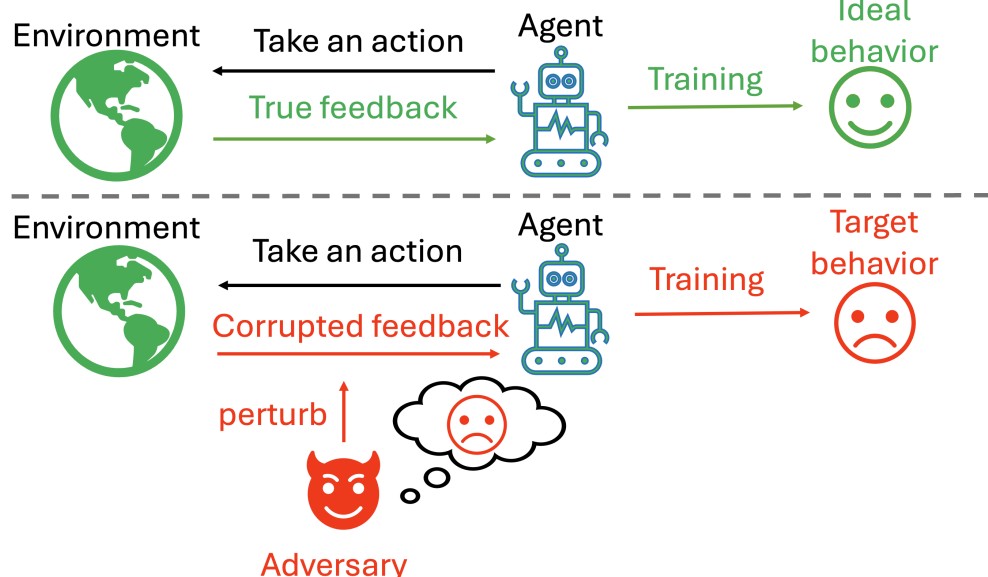

Figure 1: The framework of online reinforcement learning under adversarial poisoning attacks. The learning agent can interact with the environment by taking an action at each timestep. Ideally, when there is no attack, the agent can observe the true feedback from the environment and then learn an ideal behavior desired by the agent. However, in practice, adversaries exist in the environment that can inject corruption into the feedback from the environment. The agent can only train on the corrupted dataset and may learn a target behavior desired by the attacker instead.

## 2 Background

**Online reinforcement learning:** We consider a standard and general online reinforcement learning setting with a discounted infinite horizon (Sutton & Barto, 2018). An environment is characterized by an MDP $\mathcal{M} = (\mathcal{S}, \mathcal{A}, \mathcal{P}, \mathcal{R}, \gamma, \mu_0)$. Here $\mathcal{S}$ is the state space, $\mathcal{A}$ is the action space, $\mathcal{P} : \mathcal{S} \times \mathcal{A} \to \Delta(S)$ is the state transition function where $\Delta(S)$ is the space of state distribution, $\mathcal{R} : \mathcal{S} \times \mathcal{A} \to \mathbb{R}$ is the reward function, $\gamma$ is the discount factor, and $\mu_0$ is the distribution of the initial states. The environment is initialized at a state $s$ sampled from $s \sim \mu_0$. The agent takes an action $a$, and the environment evolves to the next state $s' \sim \mathcal{P}(s, a)$ and returns an instant reward signal $r \sim \mathcal{R}(s, a)$. At each time step, if there is no poisoning attack, the agent will receive the actual observation tuple $(s, a, s', r)$. A policy $\pi : \mathcal{S} \to \Delta(\mathcal{A})$ for the environment is a mapping from a state to a probability distribution over the action space. Here $\Delta(\mathcal{A})$ represents the space of probability distributions over the action space. Let $\mu^\pi$ be the discounted state distribution visited by a policy $\mu^\pi(s) = \mathbb{E}_{s^0 \sim \mu_0, a^\tau \sim \pi(\cdot|s^\tau)} \sum_{\tau=0}^{\infty} \gamma^\tau \cdot \mathbb{1}[s^\tau = s]$ (Agarwal et al., 2019). The performance of a policy $\pi$ is then defined as $\mathcal{J}_{\mathcal{R}}(\pi) := \mathbb{E}_{s \sim \mu^\pi, a \sim \pi(s)} \mathcal{R}(s, a)$. The optimal policy is the policy with the highest performance. Since a deterministic optimal policy always exists (Sutton & Barto, 2018), we use $\pi^*$ to denote the deterministic optimal policy and $\pi^*(s)$ to represent the optimal action at a state $s$.

**Threat model:** We consider an adversary that can inject perturbations into the reward during agent training. Let $T$ be the total number of training steps. At each time step $t$, the current state is $s^t$, and the action taken by the agent is $a^t$. The true feedback of the environment is next state $s^{t+1}$ and reward $r^t$. Before the agent observes the true feedback $(s^t, a^t, s^{t+1}, r^t)$, the adversary observes it and injects perturbation $\Delta^t$ to the reward. Then, the agent receives $(s^t, a^t, s^{t+1}, r^t + \Delta^t)$. We denote $\mathbf{\Delta} = \{\Delta^1, \dots, \Delta^T\}$ as the attacker's perturbation strategy. It is a random variable depending on the observations of the interaction between the agent and the environment.

**Constraints on the attack**: We want the attack to work in a black box setting with minimal information about the learner and environment. We formally describe the constraints of a realistic attack below.

1. **Oblivious to the learning agent & Universal attack**: The attacker is oblivious to the learning algorithms and setups of the agent. The attack should universally work on different algorithms the agents use.

2. **Oblivious to the environment**: The attacker has no knowledge of the transition function $\mathcal{P}$ and reward function $\mathcal{R}$ of the environment. It is aware of the state space $\mathcal{S}$ and action space $\mathcal{A}$.

**Budgets required by the attack**: The efficiency of the attack depends on the budgets required by the attacker. In this work, we consider the following attack budgets.

1. The total amount of the reward perturbation $||\mathbf{\Delta}||_1 = \sum_{t=1}^{T} |\Delta^t|$ during the training process the attacker applies.

2. The maximal amount of reward perturbation $||\mathbf{\Delta}||_\infty = \arg\max_{t \in [T]} |\Delta^t|$ at each step. This ensures the perturbation at each step is bounded.

**The goal of the attack**: In general, the attacker wants the agent to follow certain target behaviors during training that are characterized by a deterministic target policy $\pi^\dagger$ (Sun et al., 2020; Zhang et al., 2020). The action given by the target policy at a state is the target behavior that the attacker wants the agent to take at the state. Denote $a^t$ as the action taken by the agent under the attack and $\pi^\dagger(s^t)$ as the target action. Let $d(\cdot, \cdot) : \mathcal{A} \times \mathcal{A} \to [0, 1]$ be a distance measure between two actions. Taking the continuous action space as an example, the distance can be defined based on L2 distance as $d(a_1, a_2) = ||a_1 - a_2||_2 / L$, where $L = \max_{x,y \in \mathcal{A}} ||x - y||_2$ is the maximum 2-norm distance between any two actions. We adopt this definition for the experiments in Section 4. Note that the analysis and the method in this work apply to general distance measures. Denote $d^t = d(a^t, \pi^\dagger(s^t))$ as the distance between the agent's action and the target action at timestep $t$ and $\mathbf{d} = \{d^1, \ldots, d^T\}$ as the distances during the training process. Formally, the goal of the attack is set as follows:

The attack aims to induce a minimal difference between the actions selected by the agent and the target actions during training $||\mathbf{d}||_p^p = \sum_{t=1}^{T} (d^t)^p$ with respect to a $p$-norm.

We want to find an attack that works under the constraints and achieves the goal with a limited budget. In the next section, we formulate the attacker's goal as an optimization problem and develop efficient attack methods.

## 3 Adaptive Target Attack and Analysis

### 3.1 Reward poisoning attack as an optimization problem

Let $T$ be the total number of training steps, $\mathcal{L}$ be the class of learning algorithms from which the agent will choose its learning algorithm. Denote the state and selected action at time $t$ during training under the attack as $s^t$ and $a^t$. The perturbation at time $t$ is denoted as $\Delta^t$, which is a random variable determined by the attack strategy depending on the previous observation $\{s^{1:t}, a^{1:t}, r^{1:t-1}, \Delta^{1:t-1}\}$ during training. The attack strategy is represented by the perturbations $\mathbf{\Delta} = \{\Delta^1, \ldots, \Delta^T\}$.

We focus on universal black box attacks. The attack should be efficient against any algorithm $\text{Alg} \in \mathcal{L}$ that the agent might use without knowing exactly which algorithm is used. Ideally, an efficient attack should minimize: 1. the distance $\epsilon$ between actions taken by the agent with any $\text{Alg} \in \mathcal{L}$ and the target actions 2. the total perturbation $C$, and 3. the maximum perturbation at each step $B$. However, such an optimal attack might not exist as an attack could induce a smaller value of $\epsilon$ with more budget on $B$ and $C$. Therefore, we define the most efficient attack for a fixed perturbation budget $B$, $C$ in Eq 1.

$$\min_{\Delta^{t}=1,\ldots,T} \epsilon, \text{ s.t. } \forall \text{Alg} \in \mathcal{L},$$
$$\mathbb{E}_{\text{Alg},\mathcal{M}}[||\mathbf{d}||_p^p] \le \epsilon, \tag{1}$$
$$\mathbb{E}_{\text{Alg},\mathcal{M}}[||\mathbf{\Delta}||_1] \le C, \mathbb{E}_{\text{Alg},\mathcal{M}}[||\mathbf{\Delta}||_\infty] \le B$$

To complete the definition of the attack problem, we need to define $\mathcal{L}$, the class of algorithms the agent might use. First we argue that letting $\mathcal{L}$ include all possible algorithms makes the problem trivial and is meaningless for the attacker. In this case, there always exists an algorithm that takes specific actions regardless of the environment and the attack to maximize $\epsilon$. For example, the target policy is always taking the action in the center of the action space (suppose the action space is a multi-dimensional box), and the learning algorithm is to always take actions on the corners of the action space. Such an algorithm dominates the value of $\epsilon$ and makes it the same for all attacks. As a result, any attack that satisfies the budget constraint will be an optimal solution, which is meaningless. It is also pointless for the attacker to consider attacking an inefficient and not data-driven algorithm that any rational agent will not use. Therefore, it is more meaningful for the attacker to consider agents using 'efficient' algorithms. To quantify which kind of algorithms are 'efficient,' we generalize a common feature shared by existing efficient DRL methods in the definition below.

**Definition 3.1.** ($\delta_p$-efficient learning algorithm) Let the total training steps be $T$ and the optimal policy of the environment be $\pi^*$. A $\delta_p$-efficient learning algorithm guarantees that, in expectation, it will take the actions close to the optimal actions for most of the time during training: $\mathbb{E} \sum_{t=1}^{T} d(a^t, \pi^*(s^t))^p \leq \delta_p$, where $\delta_p > 0$ is a constant, and $d(a^t, \pi^*(s^t))$ is the distance between the agent's action $s^t$ and the optimal action of the environment $\pi^*(s^t)$ at time $t$.

In general, an efficient learning algorithm should be capable of identifying a majority of optimal actions or actions close to them within $T$ steps in expectation. During training, it will take actions that are close to the optimal actions most of the time. We argue that existing efficient DRL algorithms are likely to be $\delta_p$-efficient with a small value of $\delta_p$. First, the goal of designing learning algorithms is to guarantee that the learning algorithm can find the optimal actions for a large set of environments. Second, a reinforcement learning algorithm needs to balance the exploration-exploitation trade-off, making it necessary to explore the sub-optimal actions strategically. In practice, most algorithms select the empirically optimal action with the highest probability, such as $\epsilon$-greedy exploration (van Hasselt et al., 2016). So, the agent will identify the optimal actions or actions close to them as empirically best actions and correspondingly select those actions for most of the timesteps. This is indirectly supported by our empirical observations in Section 4. For the rest of this section, we consider $\mathcal{L}$ to be the set of $\delta_p$-efficient learning algorithms in Eq 1.

The optimality of a solution to the optimization problem in Eq. 1 can measure the efficiency of an attack. More specifically, Eq. 1 implies a definition for the attack efficiency as in Definition 3.2.

**Definition 3.2.** (($\epsilon, C, B$) efficient attack) Given a learning algorithms family $\mathcal{L}$ and a target policy $\pi^\dagger$, an attack is ($\epsilon, C, B$) efficient if its perturbation $\boldsymbol{\Delta}$ satisfies the following conditions:

$$
\begin{aligned}
&\forall \text{Alg} \in \mathcal{L}, \\
&\mathbb{E}_{\text{Alg},\mathcal{M}}[\|\mathbf{d}\|_p^p] \leq \epsilon, \\
&\mathbb{E}_{\text{Alg},\mathcal{M}}[\|\boldsymbol{\Delta}\|_1] \leq C, \mathbb{E}_{\text{Alg},\mathcal{M}}[\max_t \|\boldsymbol{\Delta}\|_\infty] \leq B,
\end{aligned}
\tag{2}
$$

where $\mathbf{d} = \{d(a^t, \pi^\dagger(s^t))\}_{t=1}^T$ are the distances between the target actions and the actions taken by the agent under the attack.

An attack is more efficient if it is ($\epsilon, C, B$)-efficient with a lower value of $\epsilon$ for the same value of $B$ and $C$. By Eq 1, finding the optimal attack is equivalent to finding an ($\epsilon, C, B$)-efficient attack with the minimal value of $\epsilon$ given $B$ and $C$. For the rest of the paper, we focus on Eq. 2 in the theoretical sections as it is more convenient to analyze. Next, we show that finding the efficiency of an arbitrary attack strategy is hard.

First, the attacker needs to predict the influence of the attack on all $\delta_p$-efficient algorithms from $\mathcal{L}$. There can be infinitely many $\delta_p$-efficient algorithms, and the attacker does not know exactly what they are. Second, predicting the actions taken by a specific unknown learning algorithm Alg is still hard. The attacker needs to know the learning algorithm and the environment dynamics to predict the actions the agent takes and the state transitions in the future. However, it can only observe the state transitions and reward signals during training in the black box setting. It is hard to approximate the environment accurately with such limited observations. With no further limitation on the learning algorithm, the learning strategy at different timesteps can be independent, making it impossible to predict the agent's future strategy through its behavior in history.

For a targeted attack that aims at misleading the agent to follow a specific behavior, it is unreliable to adopt an attack strategy if the agent's behavior under the attack is unpredictable. Therefore, it is crucial to identify the attacks that predictably influence the learning agent's behavior. Next, we show a class of attacks that satisfy this property and find an efficient instance from the class with minimal per-step corruption.

## 3.2 Efficient adversarial MDP attack framework

Here, we build an attack framework called 'efficient adversarial MDP attack' so that the efficiency of any attack instances in the framework can be estimated. We formally define the framework in Definition 3.3.

**Definition 3.3.** (Efficient Adversarial MDP attack) An adversarial MDP attack specifies an adversarial environment $\widehat{\mathcal{M}} = (\mathcal{S}, \mathcal{A}, \mathcal{P}, \widehat{\mathcal{R}})$, which has the same state space, action space, and state transition function as the true environment but a different reward function $\widehat{\mathcal{R}}$. The perturbation at time $t$ is $\Delta^t = \widehat{\mathcal{R}}(s^t, a^t) - \mathcal{R}(s^t, a^t)$, where $s^t$ is the state visited by the agent and $a^t$ is the action taken by the agent at time $t$. Under the attack, the agent will receive reward $\widehat{\mathcal{R}}(s^t, a^t)$ at time $t$. Therefore, the agent is actually trained in the adversarial $\widehat{\mathcal{M}}$ under the attack. In addition, an efficient adversarial MDP attack characterized by $\widehat{\mathcal{M}}$ should satisfy the conditions below for a perturbation bound $\Delta > 0$.

1. The target policy $\pi^\dagger$ is the optimal policy under the adversarial environment $\widehat{\mathcal{M}}$.

2. The differences between the adversarial and true reward functions are small for the actions whose distance to the target actions is small at any state. Given a value of $\Delta \in \mathbb{R}^+$, for all state $s \in \mathcal{S}$, $|\widehat{\mathcal{R}}(s, a) - \mathcal{R}(s, a)| \leq \Delta \cdot d(a, \pi^\dagger(s))^p$.

We denote EM as the class of all efficient adversarial MDP attacks. We denote an instance of EM as M($\Delta$), where $\Delta$ is its perturbation bound. In Theorem 3.4, we show a guarantee on the attack efficiency of an efficient adversarial MDP attack M($\Delta$) $\in$ EM.

**Theorem 3.4.** *An efficient adversarial MDP attack M($\Delta$) $\in$ EM is $(V, B, C)$-efficient by Eq. 2 with $\epsilon = \delta_p$, $B = \Delta$, and $C = \delta_p \cdot \Delta$.*

The proof for Theorem 3.4 can be found in Appendix A. Here, we explain how the framework is constructed and how its efficiency is evaluated. The framework is built to work closely with the $\delta_p$-efficient learning assumption. We note that the assumption describes the behavior of the agent learning from a static environment. Therefore, this assumption applies if the attacker makes the agent train as if in a malicious environment $\widehat{\mathcal{M}}$ obtained by perturbing the true environment $\mathcal{M}$. In this case, for any algorithm Alg $\in \mathcal{L}$, it is always true that $\mathbb{E}[\sum_{t=1}^{T} d(a^t, \hat{\pi}^*(s^t))^p] \leq \delta_p$, where $\hat{\pi}^*$ is the adversarial optimal policy on the adversarial environment. If the adversarial optimal policy is exactly the target policy $\hat{\pi}^* = \pi^\dagger$, then the attack guarantees that $\epsilon = \mathbb{E}[\sum_{t=1}^{T} (d(a^t, \hat{\pi}^*(s^t))^p)] \leq \delta_p$. To make the agent train as if in $\widehat{\mathcal{M}}$, the perturbation at time $T$ should satisfy $\Delta^t = \widehat{\mathcal{R}}(s^t, a^t) - \mathcal{R}(s^t, a^t)$. To limit the requirement budget, the attacker can limit the perturbation around the adversarial optimal policy, as the agent will mostly take those actions. The idea we adopt is to connect the perturbation applied to an action to its distance from the target action. More specifically, we require $|\widehat{\mathcal{R}}(s, a) - \mathcal{R}(s, a)| \leq \Delta \cdot d(a, \pi^\dagger(s))^p$. In this case, for any algorithm Alg $\in \mathcal{L}$, we have $B = \Delta$ directly by the attack definition. The total perturbation $C$ applied by the attack is connected to the total distance between the agent's actions and target actions: $\sum_t |\widehat{\mathcal{R}}(s^t, a^t) - \mathcal{R}(s^t, a^t)| \leq \Delta \cdot \|\mathbf{d}\|_p^p \leq \delta_p \cdot \Delta$.

The guarantees above suggest that an attack could be more efficient with a smaller value of $\Delta$. However, the value of $\Delta$ must be significant enough to make the target policy $\pi^\dagger$ optimal in the adversarial environment $\widehat{\mathcal{M}}$. Next, we aim to find this threshold and the corresponding efficient adversarial MDP attack with the minimal value of $\Delta$.

## 3.3 Efficient adversarial MDP attack with minimal $\Delta$: adaptive target attack

This section shows an instance of the efficient adversarial MDP attack framework that requires the minimal value of $\Delta$. First, we define the 'adaptive targeted attack' in Definition 3.5 below.

---

**Algorithm 1** Adaptive Target Attack Framework

---

1: **Input**: target policy $\pi^\dagger$
2: **Params**: distance measure $d$, maximal per-step perturbation $\Delta$, polynomial factor $q$
3: **for** $t = 1, 2, \ldots T$ **do**
4:     Observe environment state $s^t$, agent's action $a^t$, and reward signal $r^t$
5:     Perturb reward signal $r^t \leftarrow r^t - \Delta \cdot d(a^t, \pi^\dagger(s^t))^q$
6: **end for**

---

**Definition 3.5.** (Adaptive target attack) The adaptive target attack $\mathrm{AT}(\Delta, q)$ as described in Algorithm 1 is parameterized by the maximal reward perturbation $\Delta \in \mathbb{R}^+$, and a positive value $q \in \mathbb{R}^+$. Given a target policy $\pi^\dagger$ and a distance measure $d$, the reward perturbation at round $t$ by the adaptive target attack satisfies $\Delta^t = -\Delta \cdot d(a^t, \pi^\dagger(s^t))^q$, where $s^t$ is the state visited by the agent and $a^t$ is the action taken by the agent at time $t$.

The adaptive target attack penalizes the agent for taking an action different from the target action. The attack essentially constructs an adversarial environment with an adversarial reward function satisfying $\widehat{\mathcal{R}}(s, a) = \mathcal{R}(s, a) - \Delta \cdot d(a, \pi^\dagger(s))^q$. The computational resource required by the attack is the memory to store the target policy $\pi^\dagger$, and the computational power to calculate the target action given by $\pi^\dagger$ and its distance to the action taken by the learning agent.

Formally, in Theorem 3.6, we show that the efficient adversarial MDP attacks with the minimal value of perturbation bound $\Delta^*$ must be an adaptive target attack with $\Delta = \Delta^*$ and $q = p$. The proof can be found in the Appendix A. Intuitively, in the adversarial environment, the adaptive target attack with $q = p$ penalizes all non-targeted policies more than any other instances of efficient adversarial MDP attack with the same value of $\Delta$, making it easier to highlight the target policy.

**Theorem 3.6.** *Let EM be the class of efficient adversarial MDP attacks and $M(\Delta)$ be an instance with perturbation bound $\Delta$. Denote $\Delta^* = \min_{M(\Delta) \in EM} \Delta$ as the minimal value of $\Delta$ for an efficient adversarial MDP attack. The value of $\Delta^*$ is*

$$\Delta^* = \max_{\pi: D(\pi, \pi^\dagger) > 0} \frac{\mathcal{J}_\mathcal{R}(\pi) - \mathcal{J}_\mathcal{R}(\pi^\dagger)}{D(\pi^\dagger, \pi)},$$

*where $\mathcal{J}_\mathcal{R}(\pi)$ is the performance of a policy $\pi$ on the actual reward model $\mathcal{R}$ of the true environment, and $D(\pi_1, \pi_2) := \mathbb{E}_{s \sim \mu^{\pi_1}} d(\pi_1(s), \pi_2(s))$ represents the distance between two polices. In addition, the adaptive target attack $AT(\Delta^*, p)$ is the adversarial MDP attack with minimal perturbation bound $M(\Delta^*)$.*

In summary, Theorem 3.4 guarantees a lower bound on the attack efficiency of an adversarial MDP attack in Definition 3.3. The lower bound decreases as the perturbation bound $\Delta$ of an adversarial MDP attack decreases, suggesting that efficient adversarial MDP attacks should have low values of $\Delta$. We further find that the adversarial MDP attack with the minimal value of $\Delta$ is an adaptive target attack in Definition 3.5 with $q = p$. Therefore, we believe an adaptive target attack is efficient with a sufficient value of $\Delta$ and $q = p$. However, note that Theorem 3.4 provides a bound on the attack efficiency instead of the exact value. So, an adversarial MDP attack with a lower value of $\Delta$ does not necessarily have to be less efficient in practice. Also, under the $(\epsilon, B, C)$ definition for attack efficiency, the efficiencies of any two attacks are not directly comparable as there can be a trade-off between $\epsilon$ and the budgets $B, C$. Therefore, the versions of adaptive target attacks with $q \neq p$ might also be efficient in practice, and we empirically investigate this in the next section.

## 4 Experiments

Although the assumption that the algorithm used by the agent is an efficient learning algorithm in Definition 3.1 may not strictly hold in practice, we demonstrate in this section that our adaptive target attack is effective in the universal learning scenarios we test. In the theoretical analysis in Section 3, we study the general case where the attacker aims to minimize the total distances $\|\mathbf{d}\|_p^p$ between the agent's actions and the target

actions. For the simplicity of empirical evaluation, we focus on a natural case where $p = 1$, that is, the linear summation of the distances at each timestep $\|\mathbf{d}\|_1 = \sum_{t=1}^{T} d(a^t, \pi^\dagger(s^t))$ where $a^t$ is agent's action and $\pi^\dagger(s^t)$ is the target action. Accordingly, we set $q = 1$ for the adaptive target attack in most of our experiments, and we study the attack with different values of $q$ in the ablation study. Note that our attack applies to general RL settings. The rest of this section mainly focuses on the standard continuous control problems with continuous action spaces. In Appendix C, we show the attack setup and the empirical evaluations in the RL environments with discrete action spaces.

## 4.1 Experiment Setup

**Learning scenarios and target policies:** Here, we focus on the standard continuous robotic control problems from Mujoco (Todorov et al., 2012), including HalfCheetah, Hopper, and Walker. We verify that our attack works efficiently against the environments with discrete action spaces in Appendix C. For the learning algorithms, we consider state-of-the-art DRL algorithms, including DDPG (Lillicrap et al., 2015), TD3 (Dankwa & Zheng, 2019), and SAC (Haarnoja et al., 2018). The number of training steps is set as $6 \times 10^5$ to ensure that the algorithms can collect high rewards on average during training with no corruption.

Our attack is designed to work for universal target policies. Therefore, we adopt the idea of the random, medium, and expert policies from (Fu et al., 2020) to represent policies with diverse performances. A random policy is randomly generated and is usually associated with low performance. An expert policy is a policy of the highest performance that an efficient learning algorithm can learn. A medium policy has a performance in the middle of the above two. We construct such policies following the process in (Fu et al., 2020). Here, we mainly focus on the medium policy in the experiments and test all three types of policies in the ablation study. We believe that setting the target policy as a medium policy is relatively more realistic. A target policy usually has some behavior desired by the attacker but not desired by the agent, so its performance should be sub-optimal. However, if the performance of a target policy is comparable to a randomly generated policy, then the agent will find no improvement during training and may discard the trained policy in the end. Therefore, a medium policy is relatively more suitable as the target. In Appendix D, we show the details of the behavior of the medium policy we chose as a target for the HalfCheetah environment. In general, we show that the robot controlled by the medium policy exhibits unnatural behavior compared to that controlled by an expert policy. We further show that the agent learns the same strange behavior under the adaptive target attack with the medium policy as the target. In Appendix E, we show another example to more intuitively highlight the influence of the attack on the learning agent. More specifically, we test the case with a target policy optimized on the opposite of the original goal of the environment. We show that under our attack with this specific target policy, the agent learns to achieve high performance on the opposite of the actual environment learning goal, which is desired by the attacker but undesired by the agent.

**Baseline Attacks:** To the best of our knowledge, there is no existing baseline for black-box targeted online DRL attack. We adopt a state-of-the-art black-box online DRL attack from Xu et al. (2023) to our setting as a baseline, which we call the 'neighborhood attack.' The neighborhood attack is parameterized by $\Delta \in \mathbb{R}^+$ and $r \in \mathbb{R}^+$, and its corruption strategy is $\Delta^t = \Delta \cdot \mathbb{1}\{d(a^t, \pi^\dagger(s^t)) \geq r\}$. That is, it applies a fixed penalty when the agent takes an action far from the target action and no penalty otherwise. In addition, we consider a random attack strategy that randomly decides corruption at each training step as a naive baseline. More specifically, the random attack is parameterized by $\Delta \in \mathbb{R}^+$. At each training step, the corruption it applies is uniformly at random drawn from $[-\Delta, \Delta]$. In Appendix G, we provide an additional comparison to the white-box online DRL attack 'va2c-p' from Sun et al. (2020). We discuss the difference between their attacks and ours in Section 5

## 4.2 Efficiency of Attacks

First, we investigate the efficiency of attacks in different learning scenarios. The main goal is to verify that our adaptive target attack is universally effective against different efficient learning algorithms. For this purpose, we consider three environments: HalfCheetah (HC), Hopper (HP), and Walker (WK), learned by three algorithms: DDPG, TD3, and SAC. Here, we only consider the target policy to be a medium policy, as previously discussed. In the ablation study, we test other types of target policies. To ensure a fair comparison,

all the attacks share the same value of maximal per-step corruption $B$. That is, the corruption applied by the attacks at each step can not be greater than $B$. In this case, all the attacks set $\Delta = B$ to utilize the maximal per-step corruption budget fully. For the HalfCheetah environment, the maximal per-step corruption is set as $B = 50$; for the Hopper and Walker environments, the value is set as $B = 20$. In the ablation study, we test the attack with smaller values of $\Delta$. For the baseline neighborhood attack, to highlight its influence in misleading the agent, we test the values of $r \in [0.5, 3.0]$ and show the results where the attack requires a limited budget on $C$ and achieves low values of $\epsilon$. To evaluate the efficiency of the attacks, we measure the average corruption $C/T$ applied by the attack and the average distance $\epsilon/T$ between the agent's action and the target action over the number of steps $T$. Each experiment is repeated 10 times, and we report the average values together with the sample standard deviation. For more intuitive comparisons of different methods, we also report the range of the values and perform Mann-Whitney U tests in Appendix B.

The results are shown in Table 1. For our adaptive target attack, in all learning scenarios, it can always make the agent closely follow the target behavior (low value of $\epsilon$) with a limited budget required (low value of $C$). Note that the standard deviation of the results under the attack is also small, implying that the influence of the attack on the agent is stable. This indirectly verifies our assumption of the efficient learning algorithms and efficiency guarantees on our attack. More importantly, the attack results are consistent for all three learning algorithms we test with, suggesting that our attack is universally efficient against different efficient learning algorithms. In comparison, the random attack can hardly make the agent follow the target behavior while requiring much more budget. The neighborhood attack can also make the agent take actions much closer to the target actions compared to the case of no attack and random attack, but the effect is much less than that of the adaptive target attack. While it also requires a limited budget of $C$ compared to the random attack, it generally requires similar or more budget than the adaptive attack on average in most cases, indicating that the adaptive attack is more efficient in general. The standard deviations for $\epsilon/T$ and $C/T$ under the neighborhood attack are also large in some cases, indicating that the attack has a less stable effect compared to our attack. For a more direct and rigorous comparison, in Appendix B, we provide the range of the values of all experiment runs in Table 4 and perform Mann–Whitney U tests to directly compare the adaptive target attack and the neighborhood attack.

| Env-Alg | Clean ($\frac{\epsilon}{T}$) | **Adapt ($\frac{\epsilon}{T}$)** | Neigh ($\frac{\epsilon}{T}$) | Random ($\frac{\epsilon}{T}$) | **Adapt ($\frac{C}{T}$)** | Neigh ($\frac{C}{T}$) | Random ($\frac{C}{T}$) |
|---|---|---|---|---|---|---|---|
| HC-DDPG | 0.52(0.05) | **0.09(0.00)** | 0.22(0.12) | 0.62(0.01) | 4.52(0.23) | 10.12(8.28) | 25.00(0.01) |
| HC-TD3 | 0.51(0.07) | **0.11(0.02)** | 0.28(0.14) | 0.54(0.01) | 5.72(0.88) | 13.73(9.95) | 24.99(0.01) |
| HC-SAC | 0.52(0.02) | **0.11(0.02)** | 0.18(0.08) | 0.52(0.00) | 5.61(0.94) | 6.68(7.38) | 25.02(0.02) |
| HP-DDPG | 0.51(0.01) | **0.17(0.00)** | 0.35(0.19) | 0.52(0.00) | 3.37(0.08) | 6.16(4.73) | 9.99(0.01) |
| HP-TD3 | 0.45(0.03) | **0.17(0.00)** | 0.34(0.04) | 0.47(0.02) | 3.37(0.06) | 6.12(1.03) | 10.00(0.01) |
| HP-SAC | 0.43(0.01) | **0.17(0.00)** | 0.26(0.01) | 0.46(0.01) | 3.33(0.05) | 2.75(0.14) | 10.00(0.01) |
| WK-DDPG | 0.49(0.02) | **0.15(0.02)** | 0.22(0.03) | 0.51(0.04) | 3.05(0.31) | 3.32(1.22) | 10.01(0.01) |
| WK-TD3 | 0.54(0.07) | **0.16(0.01)** | 0.30(0.05) | 0.46(0.06) | 3.24(0.20) | 6.67(2.31) | 10.00(0.01) |
| WK-SAC | 0.41(0.05) | **0.18(0.01)** | 0.26(0.03) | 0.39(0.03) | 3.65(0.19) | 4.87(1.51) | 10.00(0.01) |

Table 1: Performance of attacks in different learning scenarios. In the 'Dataset-Alg' column, 'HC' represents HalfCheetah, 'HP' represents Hopper, and 'WK' represents Walker. For other columns, 'Adapt' represents the adaptive target attack, 'Neigh' represents the neighborhood attack, and 'Random' represents the random attack. The values in the brackets are the sample standard deviation. The bold values have the minimal value of $\epsilon/T$ in each row.

### 4.3 Ablation Study

In this section, we focus on measuring the efficiency of the attacks according to Definition 3.2 in different setups of the attack and the learning scenario. In Appendix C, we provide additional ablation studies on maximal per-step perturbation $\Delta$. We also test the case of the attack in the RL environments with discrete action spaces and the scenario where the attack have to stop applying corruption in the middle of the training.

**Different types of target policies:** Here, we verify that the adaptive target attacks remain efficient for the dataset collected by different types of policies. We take the HalfCheetah environment as an example and

test the attack with target policies being random, medium, and expert policies. We set $\Delta = 50$ as in the main experiments. We measure the efficiency of the attack and show the results in Table 2. We observe that the adaptive target attack always successfully misleads the agent to follow the target behavior of all kinds of target policies. The results also show that the difficulties of misleading the agent to different target behaviors differ. The random policy is the easiest target as the attacker can make the agent take actions very close to the target actions (low value of $\epsilon$) while using the lowest corruption budget of $C$ among all targets. In comparison, the expert policy is the most challenging.

| Alg-Target | Clean ($\frac{\epsilon}{T}$) | Adapt ($\frac{\epsilon}{T}$) | Adapt ($\frac{C}{T}$) |
|---|---|---|---|
| DDPG-Rand | 0.51(0.01) | 0.06(0.00) | 2.76(0.01) |
| TD3-Rand | 0.54(0.02) | 0.06(0.00) | 2.80(0.02) |
| SAC-Rand | 0.43(0.01) | 0.03(0.00) | 1.66(0.02) |
| DDPG-Med | 0.52(0.05) | 0.09(0.01) | 4.58(0.41) |
| TD3-Med | 0.51(0.07) | 0.10(0.02) | 5.24(0.49) |
| SAC-Med | 0.52(0.02) | 0.12(0.01) | 5.75(0.76) |
| DDPG-Exp | 0.52(0.07) | 0.15(0.02) | 7.36(1.07) |
| TD3-Exp | 0.45(0.09) | 0.17(0.01) | 8.45(0.66) |
| SAC-Exp | 0.42(0.05) | 0.14(0.04) | 6.98(1.90) |

Table 2: Performance of the adaptive target attack for different types of target policies.

**Different values of $q$:** Here, we study the efficiency of adaptive target attack when setting the hyperparameter $q$ to different values. We choose the HalfCheetah environment as an example and test the attack with $q = \{0.5, 1, 2, 4\}$, while the other setups are the same as the experiments in Section 4.2. The results are shown in Table 3. We observe a trade-off that with a higher value of $q$, the attack less less influence on the learning agent with a higher value of $\epsilon$, but the required budget on $C$ also decreases. We observe that when the value of $q$ is no greater than 1, decreasing the value of $q$ has a marginal influence on $\epsilon$, but the increment on $C$ is significant, which is undesired by the attacker. Similarly, when $q > 2$, increasing the value of $q$ has a limited influence on $C$, but the increment in $\epsilon$ is significant. Therefore, in the general case, $q \in [1, 2]$ intuitively achieves a good balance on the trade-off. If the corruption budget becomes a critical concern, then setting a high value of $q = 4$ is also a reasonable choice, as the attack can still effectively mislead the agent to its target policy.

| Alg-$p$ | Clean ($\frac{\epsilon}{T}$) | Adapt ($\frac{\epsilon}{T}$) | Adapt ($\frac{C}{T}$) |
|---|---|---|---|
| DDPG-0.5 | 0.52(0.05) | 0.10(0.01) | 13.59(0.76) |
| TD3-0.5 | 0.51(0.07) | 0.12(0.01) | 15.01(0.41) |
| SAC-0.5 | 0.52(0.02) | 0.12(0.01) | 14.39(0.93) |
| DDPG-1 | 0.52(0.05) | 0.09(0.01) | 4.58(0.41) |
| TD3-1 | 0.51(0.07) | 0.10(0.02) | 5.24(0.49) |
| SAC-1 | 0.52(0.02) | 0.12(0.01) | 5.75(0.76) |
| DDPG-2 | 0.52(0.05) | 0.11(0.01) | 1.29(0.12) |
| TD3-2 | 0.51(0.07) | 0.13(0.02) | 1.22(0.11) |
| SAC-2 | 0.52(0.02) | 0.12(0.01) | 1.45(0.30) |
| DDPG-4 | 0.52(0.05) | 0.21(0.02) | 0.69(0.09) |
| TD3-4 | 0.51(0.07) | 0.19(0.02) | 0.49(0.07) |
| SAC-4 | 0.52(0.02) | 0.22(0.01) | 0.60(0.06) |

Table 3: Performance of the adaptive target attack for different $p$-norm.

In this work, we do not focus on the agent's ability to detect abnormal data, which can be used to defend against data corruption. To the best of our knowledge, it remains unknown how to detect abnormal data induced by a poisoning attack during training time in an online RL setting. In Appendix F, we discuss anomaly detection at training time in RL in detail.

## 5   Related Work

**Data poisoning attack in simpler reinforcement learning** Data poisoning attacks in reinforcement learning have been considered in simpler bandit cases (Jun et al., 2018; Garcelon et al., 2020; Liu & Shroff, 2019) and tabular MDP cases (Rakhsha et al., 2020; Xu et al., 2021; Zhang et al., 2020; Banihashem et al., 2022; Liu & Lai, 2021; Rakhsha et al., 2021; Li et al., 2024). Since the deep MDP case has a much more complicated environment to learn from, the attacks mentioned here cannot be applied to the deep MDP case. Banihashem et al. (2022); Rakhsha et al. (2020) adopt the same basic attack framework as our work. Besides the difference in the definition of the attack's budget, the main limitation in their works is that their methods are only defined on discrete state and action spaces and require full knowledge of the environment. The attack problems proposed in the works are also impossible to solve in the deep MDP case due to the complexity of the environment.

**Observation perturbation attack in deep reinforcement learning** A line of work study observation perturbation attack and defense in DRL (Behzadan & Munir, 2017a;b; Inkawhich et al., 2019; Liang et al., 2022; Liu et al., 2022; Qiaoben et al., 2024). There are three main differences between these works and ours. First, observation perturbation attacks perturb the state signal, while our attack perturbs the reward signal. Second, observation perturbation attacks do not change the environment's dynamics but change the environment's signal observed by the agent. Our attack changes the dynamics of the environment. Third, the observation perturbation attack focuses on the intrinsic generalization vulnerabilities of the neural networks used by the learning agent. In contrast, our attack focuses on the vulnerabilities of the exploration strategy of the learning agent.

**Data poisoning attack in deep reinforcement learning** Xu et al. (2023) study untargeted reward poisoning attacks against DRL. Sun et al. (2020) is the only other work considering targeted attacks in DRL. There are three main limitations of this attack compared to ours and therefore less practical: (a) the attack is white-box that requires the knowledge of the learning algorithm, (b) the attack only works for on-policy learning algorithms, and (c) the attacker decides the perturbation after a batch of training steps. In addition, their attack problem is designed to minimize the cost of corruption per training batch rather than each training step. Our attack is designed to minimize the per-step corruption and total corruption, making our attack more general. For empirical evaluation, they only cover naive target policies that are not representative of practical attack scenarios. In our work, we experiment with multiple non-trivial types of target policies.

## 6   Conclusion and Limitations

In this work, we propose the first black-box targeted reward poisoning attack against online DRL. Our attack is practical because (1) it does not have knowledge of the learning agent and environment dynamics, (2) it requires a limited budget for corruption over each step and the whole training process, and (3) it requires limited computational resources. The limitations of our attack include: (1) it only studies reward poisoning attacks, (2) it only works for online reinforcement learning settings. We hope our study will motivate the development of DRL algorithms robust against data poisoning attacks in the future.

### Acknowledgments

We sincerely thank the anonymous reviewers and action editor for their insightful comments and constructive suggestions. This work was supported by funding through NSF Grants No. CCF-2238079, CCF-2316233, CNS-2148583 and a Research Gift from Amazon AGI Labs.

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

## A   Proof

**Proof for Theorem 3.4**

*Proof.* By the first property of an efficient adversarial MDP attack, the target policy is also the optimal policy on the adversarial environment constructed by the attack $\hat{\pi}^* = \pi^\dagger$. For any $\delta_p$-efficient learning algorithms, we have $\mathbb{E}[\sum_{t=1}^T d(a^t, \pi^\dagger(s^t)^p] = \mathbb{E}[\sum_{t=1}^T d(a^t, \hat{\pi}^*(s^t)^p] \leq \delta_p$.

Since $d(\cdot, \cdot) \leq 1$ is always true, the perturbation at each step satisfies $\Delta^t \leq \Delta \cdot d(a^t, \pi^\dagger(s^t)^p \leq \Delta$.

By the second property of an efficient adversarial MDP attack, the total amount of perturbation is bounded by

$$\mathbb{E}[\sum_{t=1}^T \Delta^t] \leq \Delta \cdot \mathbb{E}[\sum_{t=1}^T d(a^t, \pi^\dagger(s^t)^p] \leq \Delta \cdot \delta_p.$$

In summary, an efficient adversarial MDP attack satisfies $\mathbb{E}[\sum_{t=1}^T d(a^t, \pi^\dagger(s^t)^p] \leq \delta_p$, $\max_t \Delta^t \leq \Delta$ and $\mathbb{E}[\sum_{t=1}^T \Delta^t] \leq \Delta \cdot \delta_p$. Therefore, by the definition in Eq. 2, the attack is (at least) $(\epsilon, C, B)$-efficient with $\epsilon = \delta_p$, $B = \Delta$, and $C = \Delta \cdot \delta_p$.

$\square$

**Proof for Theorem 3.6**

*Proof.* Recall the performance of a policy $\pi$ is $\mathcal{J}_\mathcal{R}(\pi) = \mathbb{E}_{s \sim \mu^\pi} \mathcal{R}(s, \pi(s))$. The performance of a policy $\pi$ in the adversarial environment constructed by an efficient adversarial MDP attack $M(\Delta)$ can be decomposed as

$$\begin{aligned}
\mathcal{J}_{\widehat{\mathcal{R}}}(\pi) &= \mathbb{E}_{s \sim \mu^\pi} \widehat{\mathcal{R}}(s, \pi(s)) \\
&= \mathcal{J}_\mathcal{R}(\pi) - \mathbb{E}_{s \sim \mu^\pi}(\mathcal{R}(s, \pi(s)) - \widehat{\mathcal{R}}(s, \pi(s))) \\
&\geq \mathcal{J}_\mathcal{R}(\pi) - \Delta \cdot \mathbb{E}_{s \sim \mu^\pi} d(\pi(s), \pi^\dagger(s))^p \\
&:= \mathcal{J}_\mathcal{R}(\pi)(s) - \Delta \cdot D(\pi, \pi^\dagger)
\end{aligned} \tag{3}$$

Here we denote $D(\pi_1, \pi_2) = \mathbb{E}_{s \sim \mu^{\pi_1}} d(\pi_1(s), \pi_2(s))$. Note that if $D_s(\pi_1, \pi_2) = 0$, then $\pi_1$, $\pi_2$ always take the same actions at the states they can visit in an episode, which makes the two policies equivalent in effect. The inequality in the second last line can only take equality if the attack $M(\Delta)$ satisfies $\Delta^t = \Delta \cdot d(\pi(s^t), \pi^\dagger(s^t))^p$, which is also the case of adaptive target attack of $q = p$. Note that the rewards for the target actions are the same in both true and adversarial environments, so we have $\mathcal{J}_{\widehat{\mathcal{R}}}(\pi^\dagger) = \mathcal{J}_\mathcal{R}(\pi^\dagger)$.

Based on the decomposition above, the difference between the performance in $\widehat{\mathcal{R}}$ of the target policy and any policy $\pi$ with $D(\pi, \pi^\dagger) > 0$ at a state $s$ satisfies

$$\mathcal{J}_{\widehat{\mathcal{R}}}(\pi) - \mathcal{J}_{\widehat{\mathcal{R}}}(\pi^\dagger) \geq \mathcal{J}_\mathcal{R}(\pi) - \mathcal{J}_\mathcal{R}(\pi^\dagger) - \Delta \cdot D(\pi, \pi^\dagger).$$

The inequality holds if and only if the attack $M(\Delta)$ satisfies $\Delta^t = \Delta \cdot d(\pi(s^t), \pi^\dagger(s^t))^p$. Next, we define a per-step perturbation threshold as below:

$$\Delta^* := \max_{\pi : D(\pi, \pi^\dagger) > 0} \frac{(\mathcal{J}_\mathcal{R}(\pi) - \mathcal{J}_\mathcal{R}(\pi^\dagger))}{D(\pi, \pi^\dagger)}.$$

For any value of $\Delta < \Delta^*$, there exists a policy $\pi = \arg\max_{\pi : D(\pi, \pi^\dagger) > 0} \frac{(\mathcal{J}_\mathcal{R}(\pi) - \mathcal{J}_\mathcal{R}(\pi^\dagger))}{D(\pi, \pi^\dagger)}$ such that $\mathcal{J}_{\widehat{\mathcal{R}}}(\pi) - \mathcal{J}_{\widehat{\mathcal{R}}}(\pi^\dagger) > 0$. Therefore, the target policy is not the optimal policy in the adversarial environment, and

the attack cannot be an efficient adversarial MDP attack. When $\Delta = \Delta^*$ and the attack sets $\Delta^t = \Delta \cdot d(\pi(s^t), \pi^\dagger(s^t))^p$, for any policy $\pi$, we have

$$\mathcal{J}_{\widehat{\mathcal{R}}}(\pi) - \mathcal{J}_{\widehat{\mathcal{R}}}(\pi^\dagger) = \mathcal{J}_{\mathcal{R}}(\pi) - \mathcal{J}_{\mathcal{R}}(\pi^\dagger) - \Delta^* \cdot D(\pi, \pi^\dagger) \le 0.$$

Therefore, the target policy is the optimal policy in the adversarial environment in this case, which is constructed by an adaptive target attack with the same value of $\Delta = \Delta^*$ and $q = p$. Such an attack is an adversarial MDP attack, and the minimal value of $\Delta$ is $\Delta^*$ for an adversarial MDP attack. For any value of $\Delta > \Delta^*$, the adaptive target attack has $\mathcal{J}_{\widehat{\mathcal{R}}}(\pi) - \mathcal{J}_{\widehat{\mathcal{R}}}(\pi^\dagger) < 0$ for any policy different from the target policy $\pi : D(\pi, \pi^\dagger) > 0$, which means the target attack is the unique optimal policy in the adversarial environment. $\qquad\square$

## B  Training Results Details

**Training log:** To intuitively illustrate the agent's behavior during training under attacks, we provide training logs for the average distance between selected actions and target actions at each training epoch. In Fig. 2, we show the results on the HalfCheetah environment. We observe that the agent gradually takes actions closer and closer to the target actions, suggesting that it gradually learns actions close to the target actions as the optimal ones. This observation agrees with our assumptions and analysis in Section 3. We note that for medium and target policies, the average distance will also decrease as time goes on. The reason could be that there are certain similarities among policies of high performance.

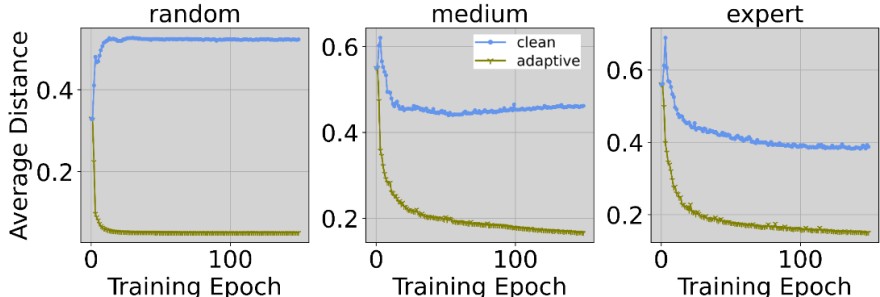

Figure 2: Training log for HalfCheetah environment under no attack and adaptive target attack with $\Delta = 50$.

**Training Result Range and Mann-Whitney U test:**

To better highlight the advantages of our method compared to the baselines, we provide the range of the values of $\epsilon/T$ and $C/T$ from all runs of each experiment in Table. 4. The results show that the range of $\epsilon/T$ under the adaptive attack is clearly disjoint from that under the random attack and no attack. This clearly demonstrates the advantage of the adaptive attack in misleading the agent to a target policy compared to the naive baselines.

For the adaptive attack and the neighborhood attack, the ranges of $\epsilon$ or $C$ under the two attacks have overlap in some cases. Therefore, to more clearly and rigorously compare the efficiency of the two attacks, we perform Mann-Whitney U test. In Table 5, we explicitly mark the cases with 'disjoint' where the two ranges have no overlap. Except for the range of $C/T$ in the Hopper environment learned by the SAC algorithm, the maximal value of $\epsilon/T$ and $C/T$ under the adaptive attack are always less than the minimal value of those under the neighborhood attack. Note that in the rest of the cases, the average values of $\epsilon/T$ and $C/T$ under the adaptive target attack are always less than those under the neighborhood attack. Therefore, we show the $p$ value for the alternative hypothesis that the distribution of $\epsilon/T$ or $C/T$ under the adaptive target attack is stochastically less than that under the neighborhood attack. For the values of $\epsilon/T$, the two ranges are disjoint in most cases, and the $p$ values are very small in the rest of the cases, indicating that the advantage of our algorithm is clear in misleading the agent into taking target actions. For the values of $C/T$, in 4 out of 9 cases, the two ranges are disjoint or the $p$ value is small. In 2 case, the value of $p$ is close to 1 or two ranges

are disjoint and the neighborhood attack has smaller values. In the rest 3 cases, while the average values under the adaptive target attacks are smaller, the $p$ values are not low. In summary, the Mann-Whitney tests show that, using a similar or smaller budget of $C$, the adaptive target attack can better mislead the agent to take actions close to the target actions compared to the neighborhood attack.

| Env-Alg | Clean ($\frac{\epsilon}{T}$) | **Adapt** ($\frac{\epsilon}{T}$) | Neigh ($\frac{\epsilon}{T}$) | Random ($\frac{\epsilon}{T}$) | **Adapt** ($\frac{C}{T}$) | Neigh ($\frac{C}{T}$) | Random ($\frac{C}{T}$) |
|---|---|---|---|---|---|---|---|
| HC-DDPG | $[0.44, 0.59]$ | $[\mathbf{0.08}, \mathbf{0.10}]$ | $[0.10, 0.49]$ | $[0.62, 0.63]$ | $[4.23, 4.96]$ | $[2.70, 25.94]$ | $[24.98, 25.01]$ |
| HC-TD3 | $[0.37, 0.58]$ | $[\mathbf{0.09}, \mathbf{0.14}]$ | $[0.10, 0.52]$ | $[0.53, 0.56]$ | $[4.44, 6.89]$ | $[2.79, 25.79]$ | $[24.98, 25.04]$ |
| HC-SAC | $[0.49, 0.54]$ | $[\mathbf{0.08}, \mathbf{0.15}]$ | $[0.11, 0.37]$ | $[0.52, 0.53]$ | $[4.05, 7.57]$ | $[1.83, 21.07]$ | $[24.98, 25.04]$ |
| HP-DDPG | $[0.49, 0.52]$ | $[\mathbf{0.17}, \mathbf{0.18}]$ | $[0.25, 0.89]$ | $[0.52, 0.53]$ | $[3.30, 3.59]$ | $[3.73, 19.96]$ | $[9.99, 10.00]$ |
| HP-TD3 | $[0.41, 0.49]$ | $[\mathbf{0.17}, \mathbf{0.18}]$ | $[0.27, 0.41]$ | $[0.44, 0.49]$ | $[3.32, 3.52]$ | $[4.43, 7.72]$ | $[9.99, 10.00]$ |
| HP-SAC | $[0.42, 0.45]$ | $[\mathbf{0.16}, \mathbf{0.17}]$ | $[0.25, 0.27]$ | $[0.44, 0.48]$ | $[3.28, 3.46]$ | $[2.58, 3.03]$ | $[9.99, 10.01]$ |
| WK-DDPG | $[0.47, 0.51]$ | $[\mathbf{0.13}, \mathbf{0.17}]$ | $[0.19, 0.28]$ | $[0.48, 0.59]$ | $[2.55, 3.44]$ | $[2.11, 6.36]$ | $[10.00, 10.02]$ |
| WK-TD3 | $[0.46, 0.65]$ | $[\mathbf{0.14}, \mathbf{0.17}]$ | $[0.24, 0.36]$ | $[0.37, 0.53]$ | $[2.81, 3.42]$ | $[3.75, 10.10]$ | $[9.99, 10.01]$ |
| WK-SAC | $[0.35, 0.49]$ | $[\mathbf{0.15}, \mathbf{0.19}]$ | $[0.19, 0.29]$ | $[0.35, 0.43]$ | $[3.09, 3.79]$ | $[1.00, 6.65]$ | $[9.99, 10.01]$ |

Table 4: Range of the values over all runs of the experiments in Table 1.

|  | DDPG | TD3 | SAC |
|---|---|---|---|
| $\epsilon/T$-HalfCheetah | disjoint | 0.001 | 0.002 |
| $\epsilon/T$-Hopper | disjoint | disjoint | disjoint |
| $\epsilon/T$-Walker | disjoint | disjoint | disjoint |
| $C/T$-HalfCheetah | 0.43 | 0.24 | 0.94 |
| $C/T$-Hopper | disjoint | disjoint | disjoint* |
| $C/T$-Walker | 0.69 | disjoint | 0.001 |

Table 5: A direct comparison between the adaptive target attack and the neighborhood attack. The names of columns are the learning algorithms, and the names of the rows are the environments and the value to compare. We use 'disjoint' to highlight the cases where the ranges of the values under the two attacks are disjoint. In most cases, the maximal value of $\epsilon/T$ or $C/T$ under the adaptive target attack is less than the minimal value of that under the neighborhood attack. The only exception is marked by 'disjoint*.' In the cases where the two ranges are not disjoint, we show the $p$ value for the alternative hypothesis that the distribution of $\epsilon/T$ or $C/T$ under the adaptive target attack is stochastically less than that under the neighborhood attack. Note that The average values of $\epsilon/T$ and $C/T$ under the adaptive target attack are always less than those under the neighborhood attack, except for the average values of $C/T$ in the learning scenario of Hopper environment learned by the SAC algorithm.

## C Additional Ablation Study

**Different values of per-step corruption $\Delta$**

Here, we study the influence of the per-step corruption $\Delta$ on the adaptive target attack. We take the HalfCheetah environment and medium target policy as an example and test the attack with $\Delta = 10, 30, 50$. The results from Table 6 show that, with a higher value of $\Delta$, the attacker can better mislead the agent to take actions close to the target actions, but it will also require the attacker to spend more corruption in total. Such a trade-off implies that increasing the value of $\Delta$ may not make the attack strictly more efficient. When the value of $\Delta$ is too small, the attack is not effective. The reason is that the per-step corruption is too small to make the target policy optimal in the adversarial reward function. However, when $\Delta$ is big enough, increasing the value of $\Delta$ does not necessarily make the attack more efficient. When $\Delta$ is large enough, the agent will gradually converge to always follow the target behavior. A higher value of $\Delta$ may not make the convergence process much faster, but it will require much more budget.

**RL environments with discrete action spaces** Here, we empirically verify that the adaptive target attack is efficient against RL environments that have discrete action spaces. In this case, we define the distance

| Alg-$\Delta$ | Clean $(\frac{\epsilon}{T})$ | Adapt $(\frac{\epsilon}{T})$ | Adapt $(\frac{C}{T})$ |
|---|---|---|---|
| DDPG-50 | 0.52(0.05) | 0.09(0.01) | 4.58(0.41) |
| TD3-50 | 0.51(0.07) | 0.10(0.02) | 5.24(0.49) |
| SAC-50 | 0.52(0.02) | 0.12(0.01) | 5.75(0.76) |
| DDPG-30 | 0.52(0.05) | 0.09(0.01) | 2.83(0.21) |
| TD3-30 | 0.51(0.07) | 0.10(0.01) | 3.01(0.26) |
| SAC-30 | 0.52(0.02) | 0.11(0.01) | 3.29(0.27) |
| DDPG-10 | 0.52(0.05) | 0.16(0.03) | 1.61(0.27) |
| TD3-10 | 0.51(0.07) | 0.16(0.03) | 1.61(0.29) |
| SAC-10 | 0.52(0.02) | 0.16(0.02) | 1.62(0.20) |

Table 6: Performance of the adaptive target attack for different values of $\Delta$.

measure between two actions as: $d(a_1, a_2) = \mathbb{1}\{a_1 \neq a_2\}$. Under this measure, the distance between two actions is 0 if they are different and 1 if they are identical. we consider two popular classical control problem from Gym (Brockman et al., 2016): MountainCar and Acrobot. For the learning algorithms, we consider the state-of-the-art algorithm: double dueling DQN van Hasselt et al. (2016). For the target policies, we consider three different types of policies, including random, medium, and expert policies. The results are shown in Table C. Similar to the case of continuous action spaces, the attack can make the agent mostly take exactly the target actions during training with very limited attack budgets. The empirical observation verifies that the adaptive target attack is also efficient against the RL environments with discrete action space.

| Env-Tar-Alg-$\Delta$ | $\epsilon$ (clean) | $\epsilon$ (adaptive) | $C$ (adaptive) |
|---|---|---|---|
| Acrobot-rand-double-5 | 0.68(0.01) | 0.02(0.01) | 0.10(0.05) |
| Acrobot-med-double-5 | 0.36(0.01) | 0.08(0.01) | 0.40(0.05) |
| Acrobot-exp-double-5 | 0.33(0.01) | 0.09(0.01) | 0.45(0.05) |
| MountainCar-rand-double-5 | 0.64(0.01) | 0.03(0.005) | 0.15(0.02) |
| MountainCar-med-double-5 | 0.33(0.01) | 0.07(0.02) | 0.35(0.10) |
| MountainCar-exp-double-5 | 0.23(0.01) | 0.07(0.02) | 0.35(0.10) |

Table 7: Efficiency of the adaptive target attack in the RL environments with discrete action spaces.

**Attack under a hard limit on total budget $C$**

Here, we consider another realistic attack scenario where the attack has to stop applying perturbation after the total perturbation reaches $C$. For the HalfCheetah environment, we set $C/T = 4$; for the Walker and Hopper environments, we set $C/T = 3$. This setup ensures that the attacks we consider will likely run out of budget during the training process. The results are shown in Table 8. Recall our analysis in Section 3 states that with a sufficient value of $B$, the adaptive attack can achieve small values of $\epsilon$ and only requires a small budget of $C$. Here, our empirical results further show that even when the attack stops applying corruption in the middle of training, our adaptive attack can still make the agent mostly select target actions. Compared to the case when there is no hard limit on $C$, the average values of $\epsilon/T$ are only slightly increased. The reason is that our attack already makes the agent learn most of the target actions as the optimal actions before running out of budget. So when there is no more perturbation, the agent will only seldom take actions far from the target actions, and it will take a long time for the agent to gather enough unperturbed new information to reverse the impact of previous corrupted information on the actions far from target actions.

## D   Robot Behavior Controlled by Different Policies

In this section, we show the simulation results of a robot controlled by different policies in the HalfCheetah environment. A detailed explanation of the environment can be found in Todorov et al. (2012); Wawrzyński (2009). In general, the cat-like robot (cheetah) is controlled to run on a flat terrain. The agent can apply torques at the conjunctions of the cheetah, and the goal is to make the cheetah run as fast as possible.

| Dataset-Alg | Clean ($\frac{\epsilon}{T}$) | Adapt (constrained) | Adapt (unconstrained) |
|---|---|---|---|
| HC-DDPG | 0.52(0.05) | 0.11(0.03) | 0.09(0.00) |
| HC-TD3 | 0.51(0.07) | 0.12(0.03) | 0.11(0.02) |
| HC-SAC | 0.52(0.02) | 0.16(0.03) | 0.11(0.02) |
| HP-DDPG | 0.51(0.01) | 0.19(0.00) | 0.17(0.00) |
| HP-TD3 | 0.45(0.03) | 0.18(0.00) | 0.17(0.00) |
| HP-SAC | 0.43(0.01) | 0.17(0.00) | 0.17(0.00) |
| WK-DDPG | 0.49(0.02) | 0.15(0.01) | 0.15(0.02) |
| WK-TD3 | 0.54(0.07) | 0.18(0.01) | 0.16(0.01) |
| WK-SAC | 0.41(0.05) | 0.20(0.00) | 0.18(0.01) |

Table 8: Performance of attacks in the learning scenarios with a hard limit on budget $C$ (constrained) compared to the scenario with no hard limit (unconstrained).

First, we show the motion of the cheetah controlled by an expert-level policy. The full video clip can be found in the supplementary materials, which provides more intuitive observations. The policy is learned by training the DDPG algorithm on the actual environment for a sufficient amount of time. In Fig. 3, we show sample frames of the motion focusing on the moment when the front leg of the cheetah touches the ground. We observe that the Cheetah runs fast and elegantly (it is more direct to see from the video clip), as expected. Particularly, whenever its front tip touches the ground, the tip is usually almost vertical to the ground. After the contact, the front tip swings backward. This is similar to how a horse runs in nature.

Next, we show that the motion of the cheetah is controlled by a medium policy, which is also the target policy we use in our experiments. The policy is learned by the TD3 algorithm training on the actual environment for an insufficient time. In this case, we observe in Fig 3 that the Cheetah runs slower than the expert policy (which is also indicated by its performance lower than expert policy performance). In addition, sometimes when the front tip of the cheetah touches the ground, it is nearly horizontal to the ground. Even more, afterward, the front leg almost immediately swings forward. Intuitively, such an unnatural running behavior is very different from how Cheetahs run in the real world, and it might cause unwanted results, such as causing damage to the joints in practice.

In the end, we show the motion of the cheetah controlled by the policy learned under the attack. Specifically, we examine the policy learned in our experiment in Section 4.2, where the target policy is set as the medium policy we show above, and the attacker applies the adaptive target attack. Although the learned policy is not exactly the same as the medium policy, we observe in Fig 3 that the cheetah controlled by the learned policy exhibits the same behavior as that controlled by the medium policy. This suggests that the adaptive target attack successfully teaches the agent a dangerous behavior with a limited attack budget.

# E  Target Policies with Specific Goals

Previously, we have tested our attacks with general target policies randomly chosen based on their performance in the environment. To show the impact of the attack on the agent more intuitively, we test a case where the target policies are optimized for a specific, concrete goal that differs significantly from the original goal of the environment.

More specifically, we choose the HalfCheetah environment as an example. In this environment, the original goal is to make the cheetah run forward as fast as possible. Therefore, we set the adversarial goal to be the opposite: make the cheetah run backward as fast as possible. Such an adversarial goal has distinct behavior from the original goal, which is concrete and unwanted by the agent. To acquire target policies that can achieve a high performance on the adversarial goal, we apply the DDPG algorithm on the Halfcheetah environment with all rewards inverted during training. The original reward in the environment is the distance the cheetah moves forward in a timestep, so the inverted reward measures the distance it moves backward, which is aligned with the adversarial goal. In this case, we acquire a target policy that achieves performance as low as $-6500$ in the environment. The performance here measures the distance the cheetah moves forward during

| Expert Policy | Target Policy | Learned Policy |

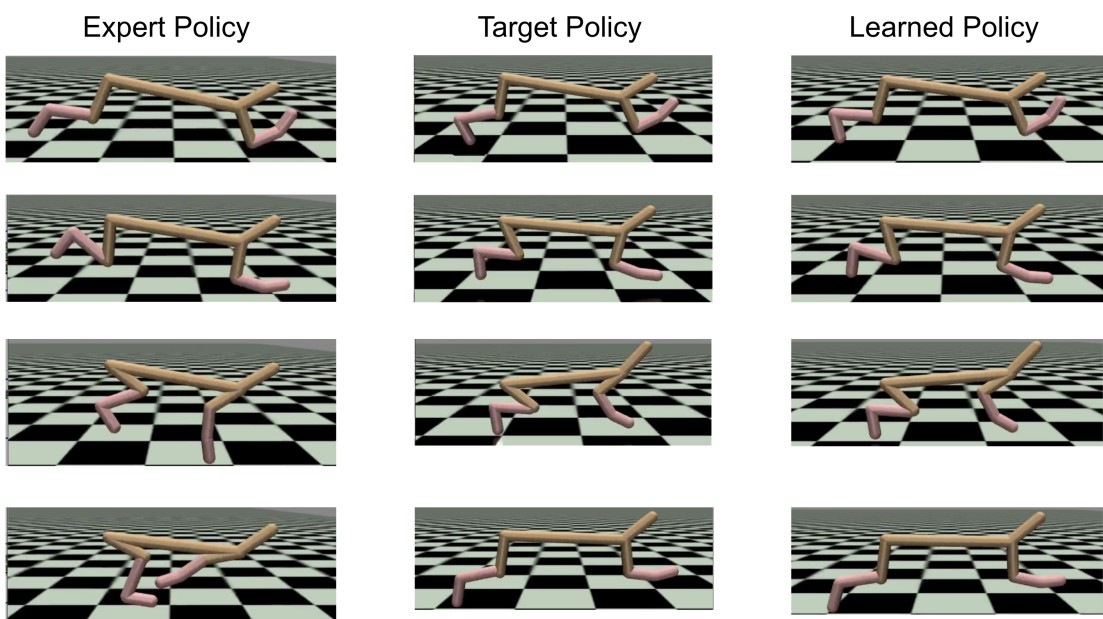

Figure 3: The motion of the cheetah when controlled by different policies. The 'expert policy' is learned by the agent when there is no attack. The 'target policy' is learned by the DDPG algorithm training for sufficient time with no attack. The target policy is the medium policy we use in our experiments in Section 4. It is learned by the TD3 algorithm training for an insufficient time. The 'learned policy' is learned by the DDPG algorithm under the adaptive target attack with the 'target policy' as the target. For each policy, we sample four frameworks before and after the front tip of the cheetah touches the ground. The contact happens between the second and the third frames.

the simulation. As a comparison, a random policy's performance is 0, and an expert policy's performance is 12000. This suggests that the target policy here makes the cheetah run backward fast.

With the policy acquired above as the target policy, we adopt the same setup as the experiments in Section 4.2 to test the performance of our attack. Our results show that the average distance between the actions taken by the agent and the target actions is $d = 0.10$, and the total budget cost is $C = 5.12$, which is similar to our observations in Table 1. The performance of the learned policy is $-5652$, which is low and similar to that of the target policy. This indicates that our attack is as efficient when the target policy is optimized on a specific adversarial goal of making the cheetah run backward. To show the results more intuitively, in the supplementary materials, we provide the video for the motion of the cheetah controlled by the target policy and the policy learned under the attack.

## F  Discussion on Attack Detectability

A practical consideration for robustness against attack is detecting the abnormality in the training data induced by poisoning attacks. We find a line of literature studying the problem of detecting the anomaly during the testing time when the learned policy is deployed in the context of RL Müller et al. (2022); Haider et al. (2023); Nasvytis et al. (2024). These works assume that the training time is uncorrupted and train their detector during the training process, which cannot be used to detect corruption during training. Therefore, it is unclear how to defend against the data poisoning attack through anomaly detection at training time in the online RLHF setting. Particularly, we find this challenging for the following reasons.

1. The distribution of the data observed in online training keeps changing during training. As a result, it is natural to see 'abnormal' data points compared to the current dataset distribution. For example, consider an agent that is initialized from a random policy and eventually converges to an expert

policy during a typical online RL training process. The distribution of the data at the beginning is collected by random behavior and has low rewards. As training continues, the agent's policy improves and starts collecting high rewards, which are 'abnormal' compared to the past reward distribution. How the data distribution changes depends on the environment and the learning algorithm, and it is unclear how it would change before training.

2. In an RL task, the agent visits different states sequentially, and at each state, it collects a reward from a reward function depending on the action it takes. In different states, the reward as a function of action can be very different. It is normal to collect high rewards at a state and low rewards at a different state using the same exploration policy.

In conclusion, it is challenging to judge if an 'abnormal' data point is induced by the attack because they naturally exist in the online RL training process.

Empirically, we test an intuitive anomaly detection method for the dynamic data distribution. The method keeps track of the most recent $N$ rewards collected during training. Let $\mu^t$ and $\sigma^t$ be the mean and variance of the past $N$ rewards before timestep $t$, the reward $r^t$ received at time $t$ is detected as abnormal if $|r^t - \mu^t| \geq \alpha \cdot \sigma^t$ where $\alpha > 0$ is a hyper-parameter set by the method. We set a moderate window size $N = 500$ and test on the Halfcheetah environment learned by the DDPG algorithm. In the case when there is no attack, in order not to detect true rewards as abnormal data, the method needs to set $\alpha > 100$. Under this setup, when the agent is under our adaptive target attack with $\Delta = 50$ (the setup in our main results), the method cannot detect any corrupted data. We have a similar observation when we set the size of the window to be different values $N = 100, 1000$. The result suggests that it can be challenging to detect data corruption by anomaly detection in the online RL setting. We believe it is a challenging yet important future work to develop efficient anomaly detection methods for online RL during training.

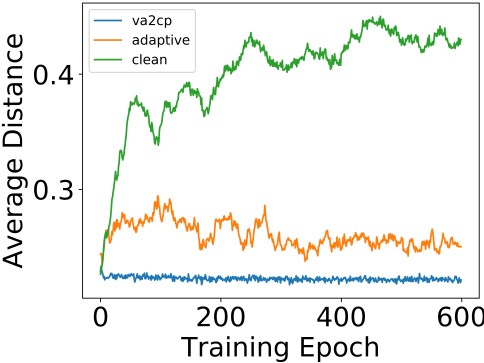

Figure 4: Training log of PPO algorithm learned in Swimmer environment under the attacks.

## G  Comparison to va2c-p

We compare our attack to the va2c-p attack from Sun et al. (2020). As an example, we consider the Swimmer environment learned by the PPO algorithm. To avoid degrading the effect of the va2c-p attack, we implement our attacks in their code and run the experiment. Due to the limitation in their code, only naive target policies are considered which always output the same action for any state. We adapt our attack to work with the constraints in their work by forcing the attack to stop applying perturbation if doing so breaks the constraint. For the values of constraints, we set $\epsilon = 0.5$ and $C/K = 1$. For our adaptive target attack, we set $\Delta = 1$. The result of the training log of the agent under the attacks is shown in Figure 4. We find that our attack has comparable efficiency to the va2c-p attack. Note that the comparison is not fair in the first place since va2c-p attack works in white-box setting while our attack works in black-box setting. Our attack is also modified here as it is optimized for a different constraint on the attack that is actually more general. We also find that our attack runs about 50 times faster than the va2c-p attack. It is noteworthy that our attack

is also much easier to implement, as its implementation is independent of the learning algorithm and only depends on the interactions during training.

