# OpenReview forum: "Universal Black-Box Targeted Reward Poisoning Attack Against Online Deep Reinforcement Learning"
_TMLR — Accepted by TMLR_

### Review · Reviewer_BFz4 · 2025-07-23

**Summary Of Contributions:**

The paper introduces a  black‑box reward‑poisoning attack against online deep reinforcement learning. The key idea is to view reward corruption as constructing a new adversarial MDP in which the optimal policy is the attacker's policy of interest. By exploiting the property that modern RL algorithm spend most of the training near the optimal actions, the attacker is able to steer learning by modifying the observed rewards. The paper formalizes this observation, proposes an attack family that works in real time without knowing the learner or the environment, demonstrates efficiency guarantees, and studies the empirical performance of the approach in both discrete and continuous control benchmarks.

**Additional Comments:**

- I think there should be new notation for the specific class
of algorithms you're restricting to in section 3.1, since
if you just look at eq 1 and the definition it is
easy to miss the fact that the specific thing that has
changed is that you have restricted the class of algorithms.

- Theorem 3.4 appears to end abruptly before finishing the statement

**Audience:**

Yes

**Audience Explanation:**

The reward poisoning approach presented here has the benefit of being completely black-box and seems to be reasonably efficient

**Claims And Evidence:**

No

**Claims Explanation:**

- Invalid threat model
  - The threat model does not seem to fit into a reasonable RL framework. How would an adversary have the freedom to intercept and perturb the rewards in an arbitrary fashion? In practice it is the practitioner who designs the reward function so as to induce a desired behavior; the attacker can't directly modify the reward function so directly. Any adversary that can arbitrarily perturb the observed reward sequence would make any problem trivial: they can always ensure that the learner observes zero reward everywhere and learns nothing.

  - It seems to me that the only way an attacker could modify the observed without breaking the standard protocol would be in settings similar to the restaurant recommendation example in the introduction: the observed rewards are not changed directly but many fake rewards are created by creating fake reviews. This is not the same as being able to arbitrarily perturb each observed reward. Instead, I think the right way to model reward poisoning attacks would be to say that the adversary has the power to perturb the reward *distribution*, rather than each individually realized reward.


- The experiments are not convincing
    - Experiments are repeated for only 10 random seeds and confidence intervals
    are reported. The results suggest statistically significant results, but I find it hard
    to believe that the reporting of confidence intervals is justified in the first place with
    so few samples; how was approximate normality assessed?

    - Baselines: it's not clear to me that the implementation of the baseline from
    Xu 2023 is a fair representation of this baseline's performance: the threshold
    for triggering the perturbation is set very low in all instances tested. For reference,
    what is the average value of $d(a_t, \pi^\dagger(s^t))$ in these environments?
    If this is much higher than 2 it would explain why the baseline appears to be inefficient. I think the
    optimal instatiation of this baseline would only penalize very different actions, rather than
    almost every action.

**Requested Changes:**

- For securing my recommendation, I request that the concerns about experiments addressed
- I think the paper would be strengthened by adding more discussion/justification/examples of the threat model. It currently seems like it aligns with almost no real RL problem

---

> ### Author Response · Authors · 2025-09-06
> **Response (1)**
>
> Thank you for your constructive comments! The revision can be found in our latest submission. Below are our responses to your concerns.
>
> **Q1:** The threat model does not seem to fit into a reasonable RL framework.
>
> **A1:** We show that the standard threat model considered in this work is realistic in a related real-world example, reinforcement learning from human or AI feedback (RLHF/RLAIF) [1,2]. Nowadays, internet companies align large language models with human feedback through reinforcement learning. In a typical online RLHF or RLAIF scenario, the learning agent prepares outputs (actions) and requests human or AI feedback. In the case of human feedback, there can be a third-party data organization that provides human feedback. Since the organization has full control over the feedback, if the organization is malicious, it has the power to change the feedback arbitrarily, which is similar to the poisoning attack framework we consider. In the case of AI feedback, the feedback can be generated by an open-source LLM. Such a model may also give biased feedback, which can be considered as corrupted reward, similar to the adversarial reward we construct.
>
> In addition, we clarify that the attack problem we consider is a standard reward poisoning attack problem in general reinforcement learning [3,4,5,6]. The main goal of this work is to highlight the vulnerability of existing DRL algorithms to reward poisoning attacks in the black-box setting. So it is suitable to study the standard reward poisoning attack model for our goal.
>
> [1]: Stiennon, Nisan, et al. "Learning to summarize with human feedback." Advances in neural information processing systems 33 (2020): 3008-3021.
>
> [2]: Lee, Harrison, et al. "Rlaif: Scaling reinforcement learning from human feedback with ai feedback." (2023).
>
> [3]: Jun, Kwang-Sung, et al. "Adversarial attacks on stochastic bandits." Advances in neural information processing systems 31 (2018)
>
> [4]: Garcelon, Evrard, et al. "Adversarial attacks on linear contextual bandits." Advances in Neural Information Processing Systems 33 (2020): 14362-14373
>
> [5]: Rakhsha, Amin, et al. "Policy teaching via environment poisoning: Training-time adversarial attacks against reinforcement learning." International Conference on Machine Learning. PMLR, 2020
>
> [6]: Sun, Yanchao, Da Huo, and Furong Huang. "Vulnerability-aware poisoning mechanism for online rl with unknown dynamics." arXiv preprint arXiv:2009.00774 (2020)

---

> ### Author Response · Authors · 2025-09-06
> **Response (2)**
>
> **Q2:** Experiments are repeated for only 10 random seeds and confidence intervals are reported.
>
> **A2:** We clarify that $10$ random seeds are standard for statistical significance analysis in DRL studies. It is prevalent to use $10$ or fewer random seeds for the evaluations [1,2,3,4,5]. The empirical results on our attack method also show that under our attack, the learning results of different learning algorithms are consistent. The variation in results of the $10$ runs is small in all learning scenarios. As an example, we repeat the experiments with $10$ different random seeds on the Hopper environment learned with the DDPG algorithm. The new empirical results are $\epsilon/T=0.17$ with variance $0.00$ and $C/T=3.34$ with variance $0.04$, which are very close to the ones we show in Table 1 in the paper.
>
> [1]: Van Hasselt, Hado, Arthur Guez, and David Silver. "Deep reinforcement learning with double q-learning." Proceedings of the AAAI conference on artificial intelligence. Vol. 30. No. 1. 2016.
>
> [2]: Haarnoja, Tuomas, et al. "Soft actor-critic: Off-policy maximum entropy deep reinforcement learning with a stochastic actor." International conference on machine learning. Pmlr, 2018.
>
> [3]: Schulman, John, et al. "Proximal policy optimization algorithms." arXiv preprint arXiv:1707.06347 (2017).
>
> [4]: Cheng, Ching-An, et al. "Adversarially trained actor critic for offline reinforcement learning." International Conference on Machine Learning. PMLR, 2022.
>
> [5]: Liu, Zhihan, et al. "Provably mitigating overoptimization in rlhf: Your sft loss is implicitly an adversarial regularizer." Advances in Neural Information Processing Systems 37 (2024): 138663-138697
>
> **Q3:** I think the optimal instantiation of this baseline would only penalize very different actions, rather than almost every action.
>
> **A3:**  Thanks for the suggestions. We perform extensive experiments on the neighborhood attack with very different setups. In the revision, we report the new results of the baseline attack with the best setup we find. In terms of misleading the agent to take actions close to the target actions (value of $\epsilon$), the neighborhood attack performs better than the random attack, but still worse than our attack by a significant margin. The neighborhood requires less budget on $C$ compared to the adaptive target attack with $q=1$. Yet, on the HalfCheetah environment, the adaptive target attack with $q=4$ achieves lower values of both $\epsilon$ and $C$ by a clear margin compared to those of the neighborhood attack, suggesting that it is strictly more efficient in this scenario.
>
> **Q4:** Additional Comments
>
> **A4:** According to suggestions from Reviewer wAsU, we have modified the statement in Theorem 3.4 for clarity. The notation $\mathcal{L}$ is first introduced to represent the class of algorithms an agent might use. Later, we specify our assumption on $\mathcal{L}$ to give it a formal mathematical definition, which completes the introduction to $\mathcal{L}$. Therefore, to avoid complexity in introducing duplicated notations, we stick to the simple notation $\mathcal{L}$ in the revision.

---

> > ### Comment · Reviewer_BFz4 · 2025-09-22
> >
> > Thank you for the response. I am satisfied with the answers except for Q3: it is not sufficient to claim that 10 seeds is standard in the field. By plotting confidence intervals you are making a claim about statistical significance; if the claim is not justified it is irrelevant whether prior works have made the same mistake.
> >
> > You should verify that the data are indeed approximately normal (although I don't see how you could make this claim confidently using only 10 samples), otherwise there is no basis for making claims of significance using confidence intervals.
> >
> > If adding sufficiently many independent trials would be beyond your computational resources, then you could consider switching to a different measure which does not require approximate normality, such as tolerance intervals.
> >
> > In the worst case, if neither of these seem viable, then the confidence intervals should just be removed along with the claims of statistical significance, and the experiments could be treated as e.g. a demonstration that might not reflect reality.

---

> > > ### Author Response · Authors · 2025-09-22
> > >
> > > Thank you for your response and suggestions!
> > >
> > > In fact, for all the values in the brackets reported in Table 1-6, they are the **variance** of our results instead of confidence intervals. We made it clear in the table's caption and results descriptions, but we realize that there is a mistake in our writing, claiming that 'Each experiment is repeated 10 times, and we report the average together with the confidence intervals.'  This is a mistake and causes confusion. We have fixed it in the revision and made all statements consistent that the results we report in the Tables are the mean and variance of the empirical results over 10 different random seeds.

---

> > > > ### Comment · Reviewer_BFz4 · 2025-09-22
> > > >
> > > > Perfect, thank you

---

### Review · Reviewer_znPE · 2025-08-11

**Summary Of Contributions:**

First of all I would like to apologize to the authors and action editor for taking longer than expected to review the paper. I find the topic of the paper interesting and I wanted to go through the claims and corresponding evidence in detail and evaluate them for their accuracy and clarity.

The summary of the contribution is:

This paper studies one way to do a "targetted" black box attack to deep reinforcement learning methods, the claim of the author is that this attack (or two attacks if one considers the base attack as separate;) is

a. "efficient": satisfies budget constraint

b. does not require knowledge of the parameters of the algorithm or even the specific algorithm being used

c. works for general RL problems with both discrete and continuous action spaces and applies to diverse learning algorithms

To make this problem tractable the authors consider and construct a class of learning algorithms that the learner (one using the RL algorithm would use) might use. And their framework "attack framework called ‘efficient adversarial reward engineering’ (is developed) such that the behavior of an arbitrary efficient learning algorithm becomes predictable under any attack under the framework."

*I'll be honest I tried understanding but the last line which is one of the core contributions remain unclear to me, and as my objections would indicate part of my concerns will be addressed if this is made more clear*.

Finally the authors present experimentation in standard environments: HalfCheetah, Hopper, Walker, and Acrobot and against standard DRL algorithms: SAC, TD3, DDPG and DQN, which although might not be state-of-the-art but are certainly widely used and serve as a decent harness for numerical simulations.

I would kindly request the authors if I have missed or misread any by-and-large contributions, it would help in an objective evaluation

**Additional Comments:**

Is there any zero-game perspective on this problem that the authors can give?

**Audience:**

Yes

**Audience Explanation:**

I believe that this paper definitely contributes to the literature of poisoning attacks in reinforcement learning. Especially I imagine in the near future, we would be able to achieve truly online continual reinforcement learning, such attacks would be concerning and studying how one can do such attacks or modelling them is very relevant to designing countermeasures. Or even using these algorithms for the good when one is trying to bypass reinforcement learning of bad actors.

**Broader Impact Concerns:**

I feel that the work is relevant in studying the behaviour of reinforcement learning algorithms in the wild and adversarial attacks on them. I do not think there are any immediate impact concerns, however I would appreicate if the authors can add a mention to how such attacks can be used if the reinforcement learning is being done for good and the attacks are by adversarial attackers.

**Claims And Evidence:**

No

**Claims Explanation:**

I will try to articulate my fundamental claims-evidence mismatch of the paper.

The authors claim " Our attack is universally efficient against any efficient learning algorithm training in general RL environments and requires limited attack budgets and computational resources." (verbatim from abstract)

However I strongly believe that the assumptions that the authors operate can only imply a weaker version of this claim and should be qualified everywhere this claim is used. Even empirically I do not find any evidence for this.

Let me try to explain this.

*Simple language*

Here is what the more general claim from the abstract and the rest of the paper can imply (or atleast what I thought without going throught the paper)

I can actually get the RL algorithm to take unsafe or actions very different from the ones they would have taken.

What the theorems, assumptions and experiments support:

The RL algorithm can indeed be attacked but the perturbed action are not very different from the optimal actions the algorithm would have taken otherwise.

This questions the *generality* and *universality* of the approach.

I believe such a work is still an important step in that direction but the claims need to be made more exact (and also the clarity of the writing) so that the evidence is calibrated.

If the authors wish to keep the claims exact, they would have to provide evidence (theoretically and empirically) for:

Environments where the action taken was completely different (the cheetah running in opposite direction for example) still respecting the constraints? I imagine not therefore the attacks are not *universal* without further qualification. I know we live in a world obsessed with universality but I personally believe simple niche results are instrumental in scientific progress and there's no special pride in claiming *universaility*.

To the cheetah example: Such an unnatural running behavior could be why the cheetah runs slower and might cause damage to the robot in the long run in practice. Can we quantify the slowness of speed, and if there is any evidence to damage for the same for real-life  studies for their damage? In the figure 2 I don't see a drastic difference in the policies.

Also are there fundamental results which can show such a *universal* attack is impossible.

*Technical*

The authors assume that the learning algorithms learn near the optimal action most of the time, this is untrue in general, therefore one should make it clearn It is true when the algorithm is minimizing cummulative regret, however there too during the exploration or *learning* the algorithms can take widely different actions. There is even work which shows that adversarial learning can benefit the actual performance of the algorithm. In fact in practice one stops RL and deploys the agent once the reward curve plateus. Therefore this central assumption does not cover efficient learning algorithms in practice and one needs to be clear in their claims and specify algorithms which take the optimal actions most of the time. (This assumption is valid for bandit algorithms and recsys algorithms though so might be worth mentioning it).

Secondly Point 2 of the efficient adversarial MDP attack, "The differences between the adversarial and true reward functions are small for the actions whose distance to the target actions is small at any state." - I mean this makes intuitive sense but shouldnt this be a feature of the MDP (or Lipschitzness of its reward function) I do not understand what the authors mean by it, but I do feel this is the assumption that leads to a tractable framework but combined with the budget constraints leads to actions which are not very different.



I also have certain problems with the framework:

1. Why do the authors not consider horizon based RL? If they do not want to, this should be made clear in the claim.

2.  Further constructing an environment given a target policy is non-trivial and so it is not clear to me how one constructs the $\hat{\mathcal{R}}$.

3. What if there is a set of optimal actions, in this case the definition of the paper are invalid and so one needs to either reduce the class of algorithms or update the definitions.

4. Please explain what p and q are? I thought both of them were selected by the attacker but then $p\neq q$ makes no sense if optimality is in $p=q$. A clear exposition of this would be very helpful. I would be able to evaluate your numerical experiments once you clarify this and other queries.

I would love to engage with the authors and help them if necessary in making the claims more accurate and the evidence more convincing.

**Requested Changes:**

There many changes that I would request the authors to make, however I would like to reserve them once the discussion is over so that I can make more targetted suggestions and fundamental changes.

As a general suggestion, I would appreciate if the authors could improve clarity of their writing and mathematical rigour. Here are few non-exhaustive suggestions. Note that I know the answer to all of these questions having read and thought about the paper extensively. However ideally a scientific paper should have a clear, simple and easy to follow exposition - it is what helps make it's dissemination easier. This list is indicative of the type of changes I would request you evaluate your paper with and improve it.

1. "The performance of a policy π is then defined as JR(π) := Es∼µπ,a∼π(s)R(s, a)" -> there are many objective for reinforcement learning (infinite horizon with discounted/average cost, finite time horizon: what is your horizon? Is your method restricted to onpolicy or applicable to off-policy methods as well?)

2. Some symbols are never defined before they are used, e.g. what does T denote?  what does $\Delta$ denote? (most likely the probability simplex but there's no point in keeping the reader guessing)

3.the term "efficiency" is abused to a certain extent, please be precise in what you mean by efficiency: sample, data, computation; similarly budget - what is the budget on? is it general? Let me take another example: "We do not consider the computational cost of constructing the target policy, as it is an input to the targeted attack problem. In general, the efficiency of an attack is measured by the attack’s influence on the learning agent’s output with limited perturbations on the training process (attack budgets).".
This is the first two lines of the paragraph. The first does not naturally follow the second one. What is the point of the paragraph is not clear. There's a lot of vagueness in the second sentence - what is influence? what is the limited perturbation? is attack budgets a modifier on training process or phrase before. Again I know what you mean because i spent a minute deciphering it, but I believe you can rewrite many such sentences much much more simply. Just a suggestion: "An attack on the RL algorithm can influence its learning. We consider perturbation of the rewards by the attack to be constrainted. Then the magnitude of the shift on the RL policy to the attacker's target policy quantifies the efficiency of the attack. The more closer a attack is able to push the RL policy, the more efficient it is" (I assume you mean closeness or influence in terms of policy as it was not clear from the original writing) There are many such examples.


4. All theorem statements and definitions should be self-complete, a reader should be able to read them independently without searching for the notation. Reintroduce assumptions, notations etc.

5. Isn't the action space and state space an input to the algorithm?

6. Why are there two constraints?? Isn't one enough (the other one can be implied with appropriate value of B/C)

Other than this please ensure that each paragraph and line has a purpose. Redundancy is important but too much fluff makes the time-sensitive reader confused. Use simple small sentences.
As I said, I'll suggest more changes once we discuss the claims and evidence more clearly.

I like the organization of the numerical section!

---

> ### Author Response · Authors · 2025-09-06
> **Response (1)**
>
> Thank you for your constructive comments! The revision can be found in our latest submission. Below are our responses to your concerns.
>
> **Q1:** What the theorems, assumptions and experiments support: The RL algorithm can indeed be attacked but the perturbed action are not very different from the optimal actions the algorithm would have taken otherwise.
>
> **A1:** We clarify that our attack can mislead an RL algorithm into taking actions close to the **target actions** specified in the attack goal, instead of the optimal actions in the true environment, during training. The target actions can be very different from the optimal actions. We have revised our definitions to make it clear.
>
> Our attack is built on the assumption that the agent would mostly take actions close to the optimal actions in the true environment. Our attack constructs an adversarial environment $\hat{M}$ where the target policy $\pi^\dagger$ is the “optimal policy” $\hat{\pi}^*$ of $\hat{M}$.
>
> The perturbations applied by the attack make the feedback received by the agent appear as if they are generated by the adversarial environment $\hat{M}$. In this case, the agent is trained on the adversarial environment in effect. By the previous assumption, the agent would mostly take actions close to the ‘optimal’ policy in the adversarial environment, which is the target policy and can be very different from the actual optimal policy $\pi^*$ in the true environment $M$.
>
> **Q2:** I imagine not therefore the attacks are not universal without further qualification.
>
> **A2:** In our experiments, we test the cases where the target policy is some random policy that can take actions very different from the optimal policy in the true environment. We show in Table 3 from the paper that under our attack, the agent will take actions close to the random policy during training with a small distance on average.
>
> The universality of our attack is a natural result of considering the attack problem in a black-box setting. Since the attacker has no knowledge of the learning algorithm used by the agent, the attacker must ensure that its attack strategy works universally for different learning algorithms that the learning agent might use.
>
> **Q3:** Can we quantify the slowness of speed, and if there is any evidence to damage for the same for real-life studies for their damage?
>
> **A3:** The slowness of the robot is directly measured by the reward of the environment. The performance of a medium policy is less than that of an expert policy, so the robot runs slower when it is controlled by a medium policy. It is also much more intuitive to see the speed of a robot as well as how it moves in the video clip we provided in the supplementary materials. We clarify in the revision that the potential damage caused by the behavior is our intuitive assumption. We made this assumption because the motion of the robot under the attack looks unnatural compared to how Cheetah runs in the real world. However, it is unclear how to quantify that in the simulation.
>
> **Q4:** Also are there fundamental results which can show such a universal attack is impossible.
>
> **A4:** [1] proves that in episodic RL, attacks can be impossible when the rewards are bounded by certain values. Our work considers an unbounded reward scenario. We design attacks that apply a low perturbation at each training step.
>
> [1]: Rangi, Anshuka, et al. "Understanding the limits of poisoning attacks in episodic reinforcement learning." arXiv preprint arXiv:2208.13663 (2022).

---

> ### Author Response · Authors · 2025-09-06
> **Response (2)**
>
> **Q5:** The authors assume that the learning algorithms learn near the optimal action most of the time, this is untrue in general
>
> **A5:** In the paper, we acknowledge that the assumption we made in Definition 3.1 may not strictly hold in practice. However, the empirical results verify the effectiveness of our attack in misleading the agent and indirectly verify that the assumption is realistic when the agent is trained in the adversarial environment constructed by the attack. The attacker constructs the adversarial environment to make the target policy have a higher performance than other policies, and the agent successfully almost always takes actions close to the target actions in the adversarial environment.
>
> To better support the above statement, in the revision, we provide training logs that record the average distance between selected actions and target actions at each training epoch in Appendix B. We observe that the agent gradually takes actions closer and closer to the target actions, suggesting that it gradually learns actions close to the target actions as the optimal ones. This observation agrees with our assumptions.
>
> In practice, greedy exploration strategies are typical choices for DRL algorithms [1,2,3]. With a high probability, it takes the empirically optimal action; otherwise, it takes a random action. In this case, if the algorithm can successfully learn the optimal action during training, it would mostly take the optimal action, which is aligned with our assumption.
>
> [1]:Dankwa, Stephen, and Wenfeng Zheng. "Twin-delayed ddpg: A deep reinforcement learning technique to model a continuous movement of an intelligent robot agent." Proceedings of the 3rd international conference on vision, image and signal processing. 2019.
>
> [2]:Lillicrap, Timothy P., et al. "Continuous control with deep reinforcement learning." arXiv preprint arXiv:1509.02971 (2015).
>
> [3]:Van Hasselt, Hado, Arthur Guez, and David Silver. "Deep reinforcement learning with double q-learning." Proceedings of the AAAI conference on artificial intelligence. Vol. 30. No. 1. 2016.
>
> **Q6:**  I mean this makes intuitive sense but shouldnt this be a feature of the MDP (or Lipschitzness of its reward function) I do not understand what the authors mean by it
>
> **A6:** Here, we focus on the difference between two MDPs: the true MDP of the environment and the adversarial MDP constructed by the attack. What we explain is a rule that an adversarial MDP should follow instead of an assumption on the true MDP of the environment. As we clarified earlier, the goal of the attacker is to mislead the agent to a target policy, which can be very different from the optimal policy. The high-level idea of the attack is to modify the reward model to make the target policy become the optimal policy in the adversarial environment. To achieve this without using too much attack budget, we require the differences between the adversarial and true reward functions to be small for the actions whose distance to the target actions is small at any state.
>
> **Q7:** Why do the authors not consider horizon based RL? If they do not want to, this should be made clear in the claim.
>
> **A7:** We have made it clear in the revision that we only consider infinite-horizon RL. It is a prevalent setting considered by empirical studies on DRL [1,2,3,4]. It is straightforward to extend our attack method to the finite horizon setting using the same perturbation strategy. We believe it is not very difficult to extend the theoretical analysis to the finite horizon setting.
>
> [1]:Dankwa, Stephen, and Wenfeng Zheng. "Twin-delayed ddpg: A deep reinforcement learning technique to model a continuous movement of an intelligent robot agent." Proceedings of the 3rd international conference on vision, image and signal processing. 2019.
>
> [2]:Lillicrap, Timothy P., et al. "Continuous control with deep reinforcement learning." arXiv preprint arXiv:1509.02971 (2015).
>
> [3]:Van Hasselt, Hado, Arthur Guez, and David Silver. "Deep reinforcement learning with double q-learning." Proceedings of the AAAI conference on artificial intelligence. Vol. 30. No. 1. 2016.
>
> [4]: Haarnoja, Tuomas, et al. "Soft actor-critic: Off-policy maximum entropy deep reinforcement learning with a stochastic actor." International conference on machine learning. Pmlr, 2018.

---

> ### Author Response · Authors · 2025-09-06
> **Response (3)**
>
> **Q8:** Further constructing an environment given a target policy is non-trivial
>
> **A8:** We clarify that one of the main contributions of this work is proposing a method to construct an adversarial environment. We have made it clear in the revision that the adaptive target attack essentially constructs a certain adversarial environment. We show the attack in Algorithm 1 and prove the efficiency of the attack in Theorems 3.4 and 3.6.
>
> **Q9:** What if there is a set of optimal actions
>
> **A9:** Under the adaptive target attack, the agent is trained in an adversarial environment constructed by the attacker. In the adversarial environment, we show that the target policy is the unique optimal policy in the environment constructed by an adaptive target attack in the proof of Theorem 3.6 in the appendix.
>
> **Q10:** Please explain what p and q are
>
> **A10:** We clarify that $p$-norm is given in the problem definition Eq (1) and (2), while $q$ is a hyperparameter of the adaptive target attack. So the attack can only decide the value of $q$. Theorem 3.6 proves that the attack achieves the minimal requirement on $\Delta$ when it sets $q=p$. We discuss the limitations of Theorems 3.6 and 3.4 in the paragraph after introducing Theorem 3.6. There, we also explain that in practice, the adaptive target attack can be efficient with other values of $q$.
>
> **Q11:** There are many objective for reinforcement learning
>
> **A11:** In the revision, we clarify that we consider the infinite-horizon case. Our attack is designed to work for universal efficient learning algorithms as introduced in Eq (1) and (2) under the assumption in Definition 3.1.
>
> **Q12:** Some symbols are never defined before they are used
>
> **A12:** Thanks for the comment. We have added clear definitions for the symbols in the appendix.
>
> **Q13:** The term "efficiency" is abused to a certain extent, please be precise in what you mean by efficiency
>
> **A13:** Thanks for the suggestion. The two lines have different purposes that cause confusion, and the explanation of the efficiency is not straightforward. We rewrite this part in the revision accordingly.  A formal definition of “efficiency” is given in Eq.2 in the paper. In general, with the same constraint on the budget, an attack’s efficiency is measured by how close the agent takes actions to the target actions during training.
>
> **Q14:** All theorem statements and definitions should be self-complete
>
> **A14:** Thanks for the comment. We have revised the theorems and definitions in the revision.
>
> **Q15:** Isn't the action space and state space an input to the algorithm?
>
> **A15:** Yes. They are implicitly included in the target policy, which takes a state from the state space as input and stochastically outputs an action in the action space. The attack strategy can be expressed based on the target policy only. Therefore, we do not list it explicitly as an input to the algorithm.
>
> **Q16:** Why are there two constraints?
>
> **A16:** We care about both the total perturbation $C$ applied by the attack and the maximal perturbation $B$ it applies at each step. The two budgets together better characterize the attack’s perturbation on the training data. If we just specify a constraint on $C$, it can only imply a weak bound on $B \leq C$, which can be too large. For example, in the HalfCheetah experiment, we set $C=5 \times 6 \times 10^5=3 \times 10^6$. This is much larger than the value we set for $B=50$.

---

> > ### Comment · Reviewer_znPE · 2025-09-12
> >
> > I thank the authors for their response. I am satisfied by most of the response - thanks for highlighting the various parts of the paper.
> >  I would appreciate if the authors can qualify what they mean by universal (in what sense) in the abstract and where it first appears in the paper.
> > My only remaining concern is that the claims might feel-overstated however I feel once the authors make the requested change and also check once if any other claims are not overstated anywhere else - I can submit my (accept) recommendation.

---

> > > ### Author Response · Authors · 2025-09-13
> > >
> > > Thanks for your responses! In the revision, we have added explanations of universal attack in the abstract and introduction. We explain in the revised abstract that "The attack is universal in that it can effectively mislead any efficient learning algorithms to follow targeted behaviors specified by
> > > the attacker under general assumptions." We have scrutinized and revised other claims made in the paper to ensure they are not overstated.
> > >
> > > Thank you again for your thoughtful and valuable comments! They greatly improve the quality and presentation of this work.

---

### Review · Reviewer_wAsU · 2025-08-23

**Summary Of Contributions:**

- This paper presents a universal black-box targeted reward poisoning attack against online deep reinforcement learning (DRL) systems. The attack is designed to work against any efficient online reinforcement learning algorithm without requiring knowledge of the specific learning algorithm or environment dynamics—only awareness of the state and action spaces is needed.
- The attacker operates by strategically perturbing reward signals during training to manipulate a DRL agent into adopting a policy that closely approximates a specific undesirable target policy, while requiring minimal computational resources and reward perturbation budgets.
- They develop an "efficient adversarial MDP attack" framework where the attacker constructs an adversarial environment with modified rewards, making the target policy optimal within this corrupted environment. The authors define $\delta_p$-efficient learning algorithms, the class of algorithms where their attack is applicable. Briefly, these are algorithms that select actions close to optimal actions for the majority of training time and the authors suggest that existing efficient deep reinforcement algorithms are $\delta_p$-efficient learning algorithms.
- The authors propose an adaptive target attack algorithm that applies reward perturbations where the penalty imposed on the reward signal is proportional to the p-norm distance between the agent's current action and the target action (action preferred by the attacker).
- The authors test their attack on DDPG, TD3 and SAC algorithms in HalfCheetah, Hopper and Walker environments and show the robustness of their method. They evaluate their method against no corruption setting and two baseline attacks - Neigh (fixed penalty if the agent takes an action that is at a distance from the target action greater than a threshold), and random (add a randomly sampled reward corruption from a specified interval).

**Audience:**

Yes

**Audience Explanation:**

Yes, the findings of the paper would be of interest to reinforcement learning and AI safety researchers. The authors show strong results in their empirical evaluations and propose a novel attack framework which could be of interest to the community.

**Claims And Evidence:**

No

**Claims Explanation:**

**Theorem 3.4** - For an efficient adversarial MDP attack $M(\Delta) \in EM$, it satisfies equation (1) with $\epsilon \leq \delta_p$, $B = \Delta$, and $C \leq (\delta_p)^p \cdot \Delta$.

**Proof sketch for theorem 3.4**-
By the definition of distance, $d(\cdot, \cdot) \leq 1$
By the definition of adversarial MDP,

$\hat{\pi}^* = \pi^{\dagger}$

By definition 3.3 (Efficient Adversarial MDP),

$\Delta^t \leq \Delta \cdot d(a^t, \pi^{\dagger}(s^t))^p \leq \Delta$

Then,

$\sum_{t=1}^{T} \Delta^T \leq \Delta \cdot \sum_{t=1}^{T} d(a^t, \pi^{\dagger}(s^t))^p = \Delta \cdot \|\mathbf{d}\|_p^p$

Applying expectation,

$\mathbb{E}[\|\Delta\|_1] \leq  \mathbb{E}[\Delta \cdot \|\mathbf{d}\|_p^p]$

Which leads to,

$C \leq E[\Delta \cdot \|\mathbf{d}\|_p^p] \leq \Delta \cdot (\delta_p)^p$

**Issues with proof for theorem 3.4** -
- If a<=b and a<=c, it does not necessarily imply b<=c. Similarly if, $\mathbb{E}[\|\Delta\|_1] \leq C$ (by definition 3.2) and $\mathbb{E}[\|\Delta\|_1] \leq  \mathbb{E}[\Delta \cdot \|\mathbf{d}\|_p^p]$ it does not necessarily imply $C \leq E[\Delta \cdot \|\mathbf{d}\|_p^p]$.
- From definition 3.1, we know $E[\|\mathbf{d}\|_p] \leq \delta_p$. I would like to request the authors to clarify how $E[\Delta \cdot \|\mathbf{d}\|_p^p] \leq \Delta \cdot (\delta_p)^p$ was derived.
---

**Theorem 3.6.** Let $EM$ be the class of efficient adversarial MDP attacks and $M(\Delta)$ be an instance with perturbation bound $\Delta$. Denote $\Delta^* = \min_{M(\Delta) \in EM} \Delta$ as the minimal value of $\Delta$ for an efficient adversarial MDP attack. The value of $\Delta^*$ is

$\Delta^* = \max_{\pi: D(\pi, \pi^{\dagger}) > 0} \frac{\mathcal{J_{R}}(\pi) - \mathcal{J_{R}}(\pi^{\dagger})}{D(\pi^{\dagger}, \pi)}.$

**Proof sketch for theorem 3.6** -

Recall the performance of a policy $\pi$ is $\mathcal{J_{R}}(\pi) = \mathbb{E}_{s \sim \mu^\pi} \mathcal{R}(s, \pi(s))$.

$\mathcal{J_{\hat{R}}}(\pi) = \mathbb{E}_{s \sim \mu^\pi} \hat{\mathcal{R}}(s, \pi(s))$

$= \mathcal{J_{R}}(\pi) \mathcal{R}(s, \pi(s)) - \mathbb{E}_{s \sim \mu^\pi} (\mathcal{R}(s, \pi(s)) - \hat{\mathcal{R}}(s, \pi(s)))$

$\geq \mathcal{J_{R}}(\pi) - \Delta \cdot \mathbb{E}_{s \sim \mu^\pi} d(\pi(s), \pi^{\dagger}(s))^p \tag{3}$

$:= \mathcal{J_{R}}(\pi)(s) - \Delta \cdot D(\pi, \pi^{\dagger})$

$\mathcal{J_{\hat{R}}}(\pi)\ge \mathcal{J_{R}}(\pi) - \Delta \cdot D(\pi, \pi^{\dagger})$

Following the above steps, the authors detail that the value of $\Delta$ to make the target policy have a higher performance than other policies $\pi : D(\pi^{\dagger}, \pi) > 0$ satisfies:

$\Delta \geq \max_{\pi: D(\pi, \pi^{\dagger}) > 0} \frac{\mathcal{J_{R}}(\pi) - \mathcal{J_{R}}(\pi^{\dagger})}{D(\pi^{\dagger}, \pi)} =: \Delta^*.$

**Issues with proof for theorem 3.6** -
- $\mathcal{J_{R}}(\pi) \mathcal{R}(s, \pi(s)) - \mathbb{E}_{s \sim \mu^\pi} (\mathcal{R}(s, \pi(s)) - \hat{\mathcal{R}}(s, \pi(s))) $,

    should be $\mathcal{J_{R}}(\pi) - \mathbb{E}_{s \sim \mu^\pi} (\mathcal{R}(s, \pi(s)) - \hat{\mathcal{R}}(s, \pi(s)))$ in the second line.

- Given that $\mathcal{J_{\hat{R}}}(\pi)\ge \mathcal{J_{R}}(\pi) - \Delta \cdot D(\pi, \pi^{\dagger})$, we get $\Delta \ge \frac{(\mathcal{J_{R}}(\pi) - \mathcal{J_{\hat{R}}}(\pi))}{D(\pi, \pi^{\dagger})}$. I would like to request the authors to clarify and provide missing steps for how they derived the $(\mathcal{J_{R}}(\pi) - \mathcal{J_{R}}(\pi^{\dagger}))$ term instead of $(\mathcal{J_{R}}(\pi) - \mathcal{J_{\hat{R}}}(\pi))$ in their final equation.


It would be helpful if the authors could clarify the above issues raised for theorem 3.4 and 3.6.

**Requested Changes:**

- Equations 1 and 2 appear to have substantial overlap, and their repeated presentation may create unnecessary redundancy. Consider consolidating or clarifying the relationship between these equations to improve readability and flow.
- "First we argue that letting $\mathcal{L}$ include all possible algorithms makes the problem trivial and is meaningless for the attacker. In this case, there always exists an algorithm that takes specific actions regardless of the environment and the attack to maximize $\epsilon$. Such an algorithm dominates the value of $\epsilon$ and makes it the same for all attacks." - It would be helpful to the readers if the authors provided an example of such an algorithm.
- In Definition 3.1, the authors state that "it will take the actions close to the optimal actions for most of the time during training, that is, $E[∥d∥_p] ≤ δp$" However $d$ is the distance between the agent's action and the target action (action favored by the attacker) as described in section 2. This seems misleading, as "optimal actions" typically refers to actions derived from the optimal policy in the true environment, not the attacker's target policy. If this definition refers to the optimal policy in the adversarial MDP, it creates confusion since the adversarial MDP framework is introduced later in Section 3.2, making this definition difficult to understand at first reading.
- In the proof of Theorem 3.6 (Appendix A), the authors present the inequality: $V_{\tilde{\mathcal{M}}}^{\pi^{\dagger}}(s) - V_{\tilde{\mathcal{M}}}^{\pi}(s) \leq (V_{\mathcal{M}}^{\pi^{\dagger}}(s) - V_{\mathcal{M}}^{\pi}(s)) + \Delta \cdot D_s(\pi, \pi^{\dagger})$. However, the derivation of this inequality and the motivation for it is not sufficiently detailed. Could the authors provide a more detailed derivation and motivation showing how they relate the value functions across these two environments?
- Minor notation error in section 3.2 - "In this case, for any algorithm $\text{Alg} \in \mathcal{L}$, it is always true that $E\left[\sum_{t=1}^{T} \left(d(a_t, \hat{\pi}^*(s_t))^p\right)^{1/p}\right] \leq \delta_p$". I think it should be -

    $E\left[(\sum_{t=1}^{T} d(a_t, \hat{\pi}^*(s_t))^p)^{1/p}\right] \leq \delta_p$.


- Minor notation errors in appendix A - $\sum_{t=1}^{T} \Delta^T \leq \Delta \cdot \sum_{t=1}^{T} d(a^t, \pi^{\dagger}(s^t)^p = \Delta \cdot \|\mathbf{d}\|_p^p$. If I am right, It should be -

    $\sum_{t=1}^{T} \Delta^t \leq \Delta \cdot \sum_{t=1}^{T} d(a^t, \pi^{\dagger}(s^t))^p = \Delta \cdot \|\mathbf{d}\|_p^p$
- Typo in section 3.2 - "optimal in the adversarial environment $\hat{\mathcal{R}}$"
- Typo in appendix A, proof for theorem 3.6- "has a higher".
- Typo in appendix B, "We test the attack with different hard limits, C = 2.5, 5, 7.5". Should be C/T.

---

> ### Author Response · Authors · 2025-09-06
>
> Thank you for your constructive comments! The revision can be found in our latest submission. Below are our responses to your concerns.
>
> **Q1:** Issues with proof for Theorem 3.4
>
> **A1:**
> We modify Theorem 3.4 and its proof in the revision for clarity. The revised goal is to prove that the efficient adversarial MDP attack is $(V,B,C)$-efficient with $\epsilon = \delta_p$, $B=\Delta$, and $C = \delta_p \cdot \Delta$. We prove this by deriving upper bounds on the distance between agent actions and target actions, as well as the perturbations applied by the attack.
> Thanks for pointing out the typo. In definition 3.1, it should be $||d||_p^p \leq \epsilon$. The upper bound on the total perturbation in the analysis afterwards should be $\delta_p$ instead of $\delta_p^p$. We have fixed them in the revision.
>
> **Q2:** Issues with proof for Theorem 3.6
>
> **A2:**
> Thanks for pointing out the typo. We have fixed the equation and rewritten the proof with more details in the revision. Here we highlight some key clarifications that can make the proof easy to follow.
>  Since the rewards are the same for the actions given by the target policy in both $R$ and $\hat{R}$, we have $J_R(\pi^\dagger=J_{\hat{R}}(\pi^\dagger)$. Therefore, the performance gap between $\pi^\dagger$ and any policy $\pi$ on the adversarial environment satisfies $$J_{\hat{R}}(\pi^\dagger)-J_{\hat{R}}(\pi)=J_{R}(\pi^\dagger)-J_{\hat{R}}(\pi) \leq J_{\hat{R}}(\pi^\dagger) - J_{R}(\pi) + \Delta \cdot D(\pi,\pi^\dagger).$$
>
> The equality holds only when the attack satisfies $\Delta^t=\Delta\cdot d(\pi(s^t),\pi^\dagger(s^t))^p$, which is the case of the adaptive target attack.
>
> Let $\Delta^*:=\max_{\pi:D(\pi,\pi^\dagger)>0} \frac{(J_{R}(\pi)-\mathcal{J}_{R}(\pi^\dagger)}{D(\pi^\dagger,\pi)}$.
>
> Recall that for an efficient adversarial MDP, the target policy should be its optimal policy. When $\Delta < \Delta^*$, there exists a policy $\pi$ such that $J_{\hat{R}}(\pi^\dagger)-J_{\hat{R}}(\pi)<0$, so the target policy is not the optimal policy, and the attack is not an efficient adversarial MDP attack.
>
> When $\Delta < \Delta^*$ and the attack satisfies $\Delta^t=\Delta\cdot d(\pi(s^t),\pi^\dagger(s^t))^p$,
>
> we have $J_{\hat{R}}(\pi^\dagger)-J_{\hat{R}}(\pi) \geq 0$ for any policy $\pi$. Therefore, the minimal value of $\Delta$ for an efficient adversarial MDP attack is $\Delta=\Delta^*.$
>
> **Q3:** Equations 1 and 2 appear to have substantial overlap, and their repeated presentation may create unnecessary redundancy.
>
> **A3:** Thanks for the suggestion. We have added more discussion on the relation between equations 1 and 2 in the revision.
>
> **Q4:** It would be helpful to the readers if the authors provided an example of such an algorithm.
>
> **A4:** Thanks for the suggestion. We have included an example in the revision.
>
> **Q5:** If this definition refers to the optimal policy in the adversarial MDP, it creates confusion since the adversarial MDP framework is introduced later in Section 3.2, making this definition difficult to understand at first reading
>
> **A5:** Thanks for pointing out the issue. There is a typo in Definition 3.1. Here, the distance notation “$d^t=d(a^t,\pi^*(s^t))$” should be measuring the distance between the action taken by the agent and the optimal action instead of the target action. Since we already use the notation $d^t$ for measuring the distance from the target action in other definitions, we avoid using $d^t$ in Definition 3.1 in the revision.
>
> **Q6:** Other typos
>
> **A6:** Thanks for pointing out the typos. We have fixed them in the revision.

---

> ### Author Response · Authors · 2025-10-01
>
> Dear Reviewer, thanks again for your valuable comments! They greatly improve the quality and presentation of this study. Please let us know if you still have questions about the claims, evidence, and audience of this work. We are happy to provide additional evidence.

---

> > ### Comment · Reviewer_wAsU · 2025-10-06
> >
> > I would like to thank the authors for making the necessary changes to the document.
> >
> > Although my technical concerns have been addressed adequately and I am leaning towards accepting the paper, I believe the paper's impact and accessibility could be further strengthened by incorporating a more intuitive experimental demonstration. The current experiments use medium or random policies in continuous and classical control problems as target policies, where the attack's effects can be subtle and difficult to interpret. Adding an experiment in a simpler environment (e.g., a grid-world) with a more concrete and well-defined target policy would make the attack mechanism more transparent and its effects more readily observable.

---

> > > ### Author Response · Authors · 2025-10-07
> > >
> > > Thanks for your response! We are very happy that your concerns have been addressed, and we truly appreciate your acknowledgement of this work.
> > >
> > > We also thank you for the new suggestion. Note that our theoretical results show that our attack can work with any target policies, and the empirical results show that the attack is successful with various target policies in multiple RL environments. Still, we agree that additional experiment results with more concrete target policies, as suggested, can further increase the accessibility and the impact of this paper. Given that the discussion period has ended, we will make a quick response now and show our plan for the new experiments as below. We will add the results to the appendix in the revision once they are ready.
> > >
> > > To make the demonstration intuitive while impactful in DRL, we consider training new target policies that optimize specific adversarial goals on the popular DRL environments, such as HalfCheetah. We will make these goals concrete and very different from the original goal of the environment. More specifically, in the HalfCheetah environment (or other continuous control problems), the original goal is to control a bot to move forward as fast as possible. Then, an adversarial goal can be making the bot move backward as fast as possible.

---

> > > > ### Author Response · Authors · 2025-10-08
> > > >
> > > > Dear reviewer, thanks again for your comment! According to the latest suggestion, we have added the new results in Appendix E and additional experiment observations in the supplementary materials. Below is the detail.
> > > >
> > > > Basically, we consider testing on the HalfCheetah environment. As we mentioned earlier, the original goal of the environment is to make the cheetah run forward as fast as possible, and our adversarial goal is to make it run backward, which is concrete and distinct from the original goal. We train a target policy that optimizes the adversarial goal. We achieve this by training on the HalfCheetah environment with all rewards inverted. The original reward in the environment is the distance the cheetah moves forward in a timestep, so the inverted reward measures the distance it moves backward, which is aligned with the adversarial goal.
> > > >
> > > > In this case, we acquire a target policy that achieves performance as low as $-6500$ in the environment. The performance here measures the distance the cheetah moves forward during the simulation. A negative performance means the cheetah moves backward. As a comparison, a random policy's performance is around $0$, a medium policy's performance is $6000$, and an expert policy's performance is $12000$. This suggests that the target policy here makes the cheetah run backward fast.
> > > >
> > > > We test our attack with the new target policy acquired above. The experiment setup is the same as that of our experiments in Section 4.2. Our results show that the average distance between the actions taken by the agent and the target actions is $d=0.10$, and the total budget cost is $C=5.12$, which is similar to our observations in Table 1. The performance of the learned policy is $-5600$, which is low and similar to that of the target policy. This indicates that our attack is as efficient when the target policy is optimized on a specific adversarial goal of making the cheetah run backward.
> > > >
> > > > To see the results more intuitively, in the supplementary materials, we provide the videos for the motion of the cheetah controlled by the target policy and the policy learned under the attack. Both videos clearly show how the Cheetah runs backward fast when controlled by both policies.

---

> > > > > ### Comment · Reviewer_wAsU · 2025-10-15
> > > > >
> > > > > I would like to thank the authors for the thorough response and for adding the new experiment. This additional experiment provides an intuitive demonstration of the attack and addresses my previous concern. I believe this revision significantly enhances the paper's contribution. Thank you.

---

> > > > > > ### Author Response · Authors · 2025-10-16
> > > > > >
> > > > > > Thank you a lot for helping us improve the quality of this work. We sincerely appreciate your comments and suggestions!

---

### Decision · Action_Editor_yS9Q · 2025-10-15

**Recommendation:** Accept with minor revision

**Additional Comments:**

I am hoping the authors (i) can clarify whether the reported number is "sample standard deviation" or "variance", (ii) can provide all 10 sample statistics of all reported metrics of all baselines, and (iii) provide the ranges of the sample metrics rather than a summary statistics like sample standard deviation.  Even better, (iv) do a Mann-Whitney test (or something similar) to make concrete arguments with statistical clarity on quantities you claim are different.

**Audience:**

Yes

**Audience Explanation:**

No disagreement here.

**Claims And Evidence:**

No

**Claims Explanation:**

I find myself agreeing with reviewer BFz4, that the empirical results lack a convincing and clear interpretation of statistical efficacy of the claims.  I suspect that the results may be sufficient to meet the authors' claims, but it's not clear from the presentation.  The authors' initial response that 10 trials is traditionally sufficient, is not sufficient.  Unconvincing practices of others is not the basis for convincing evidence.

I'll first look at the specific issue raised and responded to.  You report "Each experiment is repeated 10 times, and we report the average results together with their variance."  I suspect that you don't mean "variance" here, but rather "standard deviation"?  Is that correct?  Maybe even more accurate to say the "sample standard deviation".

However, the primary issue is two-fold: one conceptual issue and one statistical.

First, I don't understand what the reader is supposed to conclude from Table 1.  While the text says that all approaches have a maximal per-step corruption budget, the table seems to show very large differences in actual per-time-step corruption (if I'm reading it right).  And comparing to the neighborhood attack, the adaptive target method has a considerably higher corruption, suggesting the method is just employing higher degrees of corruption to achieve its lower average distance.  How does this observation affect your conclusion?

Second, is a sample standard deviation from 10 samples sufficient to evaluate the statistical validity of the claims derived from the empirical results?  The authors I think want to point to very low standard deviation values to suggest that it is sufficient, even with such a small sample size.  This first depends critically on whether the reported numbers are variance (standard deviation squared) or standard deviation.  But also, can we expect the sample standard deviation to give a good estimate of the population standard deviation with 10 samples?  One option that could help would be to report all the observed e/T and C/T values over all 10 samples.  If all of the sampled values are significantly lower than all of a baselines sampled values, that can give statistical power (under the null hypothesis seeing 10 sampled differences have the same sign is enough to give statistical significance; see sign tests or specifically Mann-Whitney).  So, providing the min and max values in the column as an interval, rather than the standard deviation, and then reporting all sampled results might be sufficient.

I think this needs to be addressed in some fashion before it can be published.

I'm going to call this "Minor revision", as I believe that I can validate whether the changes are sufficient without sending it out for a full review cycle.

---

> ### Author Response · Authors · 2025-11-07
>
> We have uploaded the camera-ready and ensured that all the suggestions and requested changes are adequately included. We sincerely thank all the reviewers and the action editor for your constructive comments and suggestions that greatly helped us improve this work, and we formally acknowledge your contribution in the camera-ready.
>
> In addition, we thank the action editor for the latest comments and suggestions. Below, we explain the changes we made in the camera-ready to address them.
>
> 1. We have made it clear that we are reporting the sample standard deviation.
>
> 2. We have made it clear that in Table 1, we report the distance and total perturbation budget averaged over the time horizon $T$. $\Delta$ is the upper bound on the perturbation an attack can apply at each individual step. We made it clear that the attack can never apply corruption greater than $B$ ($\Delta=B$) at each step.
>
> 3. Based on the suggestion, we optimize the setup on $r$ of the neighborhood attack such that it uses a limited but decent amount of budget and achieves a significant influence on misleading the agent to take actions close to the target actions. Under our latest setup, the neighborhood attack uses much less budget $C$ than the random attack and achieves a much lower average value of $\epsilon$ than that under the random attack and no attack. Still, the average values of $\epsilon$ and $C$ are higher than those under the adaptive target attack, suggesting that our attack is more efficient in general.
>
> 4. Since there are too many experiments (117 experiments in the main text, each is repeated for $10$ runs), we find it overwhelming to show the results of all experiment runs in the paper. Therefore, we provide the **ranges** of the values of $\epsilon/T$ and $C/T$ from all runs of each experiment to better highlight the advantages of our method compared to the baselines. The results in Table 4 (in camera-ready) show that the ranges of $\epsilon/T$ and $C/T$ under the adaptive attack are disjoint from those under the random attack and no attack. This clearly demonstrates the advantage of the adaptive attack in misleading the agent to a target policy compared to the naive baselines.
>
> 5. For the adaptive attack and the neighborhood attack, the ranges of $\epsilon$ or $C$ under the two attacks have overlap in some cases. Therefore, to more clearly and rigorously compare the efficiency of the two attacks, we perform **Mann-Whitney U tests**. In Table 5 from the camera-ready, we explicitly mark the cases with `disjoint' where the two ranges have no overlap. Except for the range of $C/T$ in the Hopper environment learned by the SAC algorithm, the maximal value of $\epsilon/T$ and $C/T$ under the adaptive attack are always less than the minimal value of those under the neighborhood attack. Note that in the rest of the cases, the average values of $\epsilon/T$ and $C/T$ under the adaptive target attack are always less than those under the neighborhood attack. Therefore, we show the $p$ value for the alternative hypothesis that the distribution of $\epsilon/T$ or $C/T$ under the adaptive target attack is stochastically less than that under the neighborhood attack. For the values of $\epsilon/T$, the two ranges are disjoint in many cases, and the $p$ values are very small in the rest of the cases, indicating that the advantage of our algorithm is clear in misleading the agent into taking target actions. For the values of $C/T$, in $4$ out of $9$ cases, the two ranges are disjoint (adaptive target attack always has smaller values) or the $p$ value is small. In $2$ case, the value of $p$ is close to $1$ or the two ranges are disjoint, and the neighborhood attack has smaller values. In the rest $3$ cases, while the average values under the adaptive target attacks are smaller, the $p$ values are not low. In summary, the Mann-Whitney tests show that, in general, using a similar or smaller budget of $C$, the adaptive target attack can better mislead the agent to take actions close to the target actions.
>
> Thanks again to all reviewers and the action editor for the great discussion.

---

> > ### Comment · Action_Editor_yS9Q · 2025-11-07
> > **Review of Changes**
> >
> > Thanks for the detailed description of the changes.  I have reviewed the changes and they fully address the remaining criticisms.